# Environmental redox conditions and strain variation define phenazine-mediated antagonism in co-infecting bacteria

Katlyn Todd[1,☉], Olivia Schneider[1,☉], Joshua M. Lawrence[2,3], Josefina L. Aronoff[1], Bartosz Witek[2,3], Valerie Velázquez-Colón[1], Verónica Santana-Ufret[4], Nicole L. Anderson[1], Krista Gunter[1], Moraima Noda[1], Ryan F. Relich[1,5], Lifan Zeng[6], Dominique H. Limoli[7], Christopher Whidbey[8], Jay Vornhagen [1]*

**1** Department of Microbiology and Immunology, Indiana University School of Medicine, Indiana University, Indianapolis, Indiana, United States of America, **2** Yusuf Hamied Department of Chemistry, University of Cambridge, Cambridge, United Kingdom, **3** Department of Biochemistry, University of Cambridge, Cambridge, United Kingdom, **4** Department of Obstetrics and Gynecology, Indiana University School of Medicine, Indiana University, Indianapolis, Indiana, United States of America, **5** Department of Pathology and Laboratory Medicine, Indiana University School of Medicine, Indiana University, Indianapolis, Indiana, United States of America, **6** Chemical Genomics Core Facility, Department of Biochemistry and Molecular Biology, Indiana University School of Medicine, Indiana University, Indianapolis, Indiana, United States of America, **7** Department of Biology, Indiana University, Bloomington, Indiana, United States of America, **8** Department of Chemistry, Seattle University, Seattle, Washington, United States of America

☉ These authors contributed equally to this work.

* jayvornh@iu.edu

## Abstract

*Pseudomonas aeruginosa* and *Klebsiella pneumoniae* are gram-negative opportunistic pathogens that frequently colonize the human body and are major causes of infection. These bacteria are often co-isolated in polymicrobial urinary tract and lung infections, the latter of which is associated with increased disease severity and worse clinical outcomes. Despite their overlapping niches and clinical relevance, little is known about how these two pathogens interact and how those interactions influence human health. Given the growing recognition that microbial interactions are key drivers of disease, we investigated how *P. aeruginosa* and *K. pneumoniae* influence one another. We discovered an antagonistic interaction in which *P. aeruginosa* restricts the growth of *K. pneumoniae*. This inhibition is driven by phenazine production in *P. aeruginosa*, specifically the secondary metabolites pyocyanin and pyorubin, which are both necessary and sufficient to suppress *K. pneumoniae* growth. Using a diverse set of clinical isolates, we found that this antagonism is strain-dependent. Both the susceptibility of *K. pneumoniae* to phenazines and the ability of *P. aeruginosa* to restrict *K. pneumoniae* growth varies between strains. Moreover, the necessity of phenazine production is specific to the site of infection. Together, these findings demonstrate that strain background and environmental context are critical determinants of pathogen interactions. These findings reveal that both strain background and

**Data availability statement:** All raw data, including all CFU/mL values, and analysis scripts used for this study are available at https://github.com/jayvorn/Pseudomonas-aeruginosa-and-Klebsiella-pneumoniae-phenazines and at https://doi.org/10.5281/zenodo.19501469. All raw spectroscopy data are available on the NIH Common Fund's National Metabolomics Data Repository (http://dx.doi.org/10.21228/M8D858). Pa and Ec assemblies are available on the Sequence Read Archive (Bioproject PRJNA1311040).

**Funding:** This work was supported by funding from National Institutes of Health (https://www.nih.gov/) grants R00 AI153483 to J.V. Additionally, this research was supported in part by Lilly Endowment, Inc., through its support for the Indiana University Pervasive Technology Institute. The funders had no role in study design, data collection and analysis, publication decision, or manuscript preparation.

**Abbreviations:** CF, *cystic fibrosis*; Ec, *Escherichia coli*; HR-MS, high-resolution mass spectrometry; Ko, *Klebsiella oxytoca*; Kp, *Klebsiella pneumoniae*; MBC, minimum bactericidal concentration; MIC, minimum inhibitory concentration; Pa, *Pseudomonas aeruginosa*; PBS, phosphate-buffered saline; PCA, phenazine-1-carboxylic acid; PCN, phenazine-1-carboxamide; PYO, pyocyanin; PYR, pyorubin; QS, quorum sensing; UTIs, urinary tract infections; 1-HP, 1-hydroxyphenazine; 5MPCA, 5-methylphenazine-1-carboxylate.

environmental redox conditions govern the ecological rules of pathogen interaction, providing a framework for predicting outcomes.

## Author summary

Many microbes use small molecules to compete. We found that phenazines, which are redox-active pigments made by *Pseudomonas*, stop *Klebsiella* from growing, but only in certain environments. This is because phenazines work by changing how electrons flow in cells, which depends on oxygen and other environmental factors. We show that phenazines are both necessary and sufficient to stop *Klebsiella* from growing and that different *Klebsiella* lineages vary in their susceptibility. Our results reveal a simple rule: the environment tunes chemical warfare between microbes. This principle helps explain why the same species interact differently in different human body sites and environments. drive strain- and oxygen-dependent inhibition of, revealing context-specific rules of competition.

## Introduction

Microbial interactions are increasingly recognized as critical determinants of human health, reflected by the rapid growth of microbiome research. These interactions influence a wide range of physiological outcomes and contribute to human development and disease (conceptually reviewed in [1]). The gut is a well-studied site where trillions of viruses, bacteria, and microbial eukaryotes shape immunological development, metabolism, disease susceptibility, endocrine function, and neurological health (reviewed in [2–6], amongst many others). Beyond the gut, microbial interactions influence host biology in the oral cavity, lung, nasopharynx, respiratory tract, skin, and urinary and reproductive systems. Yet, little is known about the specific mechanisms that underpin these interactions, and how they may shape human health.

*Pseudomonas aeruginosa* (*Pa*) and *Klebsiella pneumoniae* (*Kp*) are gram-negative opportunistic pathogens that are frequent colonizers of the human body (reviewed in [7–9]). They are also important causes of similar bacterial infections, such as pneumonia, urinary tract infections (UTIs), and bacteremia. Despite occupying similar niches, little is known about how these bacteria interact. Some laboratory-based studies suggest that *Pa* can restrict *Kp* growth [10]. *In vivo* studies and clinical reports indicate that concurrent infection is associated with worse outcomes [11–14]. Commensurately, other laboratory-based studies demonstrate that *Pa* and *Kp* can co-exist in the same niche [15–17]. This niche overlap implies that interaction between these genera is possible, which is supported by evidence of genetic transfer between them [18].

Redox-active natural products mediate microbial competition across environments, and these interactions can be consequential for environmental outcomes that are dependent on the fitness of specific microbes, such as polymicrobial community structure, bacterial virulence, and infection [19–22]. One of the most important classes of redox-active natural products are the phenazines [23]. Phenazines

produced by *Pa*, including pyocyanin, phenazine-1-carboxylic acid, and pyorubin, modulate respiration, iron acquisition, and biofilms, and display broad antimicrobial activities across kingdoms. Yet how environmental context gates their efficacy in mediating microbial competition remains unclear.

Understanding the molecular interactions between *Pa* and *Kp* is important for several reasons. First, both can be highly resistant to antibiotics, which poses significant threats to public health. Antibiotic resistance greatly complicates infection treatment, leading to extreme excess healthcare costs and mortality. *Kp* and *Pa* are ranked as top ten 2024 WHO Bacterial Priority Pathogens, and both organisms are ranked in the top six most important causes of healthcare burden due to antimicrobial resistance [24,25]. Second, *Pa* and *Kp* employ distinct strategies to maximize fitness in polymicrobial environments. *Pa* is highly responsive to microbial and environmental cues, deploying quorum sensing (QS) networks (reviewed in [26]), secreted and contact-dependent antagonistic mechanisms (for example [27–29]), and forming complex biofilms that enhance environmental adaptability (reviewed in [30,31]). In contrast, *Kp* utilizes metabolic flexibility and stress resistance to persist in diverse niches (reviewed in [31,32]). Additionally, the molecular mechanisms underlying host-specific fitness likely differ between these genera (reviewed in [31]). Third, *Pa* and *Kp* are highly responsive to environmental context (for example [33–35]), and possess extensive genetic diversity, with both genera harboring open pangenomes exceeding 100,000 genes. Their pangenomes are comprised of conserved core genes and variable accessory elements which confer adaptability and strain-specific phenotypes. This genetic plasticity shapes behaviors like nutrient acquisition, resilience to environmental stress, and the deployment of antagonistic effectors (reviewed in [9,36,37]). Finally, metabolic environment greatly impacts *Kp* and *Pa* pathogenesis [38–41]. As such, studying how *Pa* and *Kp* interact is important for understanding microbial interactions in general, and how they apply to human health.

This study is built off an initial observation that mice colonized with native *Pa* strains demonstrated low *Kp* gut colonization density. Thus, we hypothesized that *Pa* directly restricts *Kp* growth. We aimed to determine (1) if *Pa* can restrict *Kp* growth, (2) if restriction was due to overlapping nutritional niches, contact-dependent killing, or production of *Kp*-restrictive secondary metabolites, and (3) what pathways are required for *Pa* restriction of *Kp* growth. The work presented here demonstrates that *Pa-Kp* competition is multifactorial but primarily driven by production of *Kp*-restrictive secondary metabolites, which is dependent both on strain and environmental context.

## Results

### *Pseudomonas aeruginosa* restricts *Klebsiella pneumoniae* growth in a contact-independent manner

In the process of performing *in vivo* experiments unrelated to this study, we observed the presence of an indigenous gut bacterium associated with a reduction in *Kp* colonization from 24 to 48 hours after oral inoculation (S1A Fig). The goal of this experiment was to assess colonization across multiple mouse vendors and barriers (specific room in which the animals are housed), both of which are known to impact the gut microbial community structure [42]. *Kp* colonization was tested in three vendor backgrounds (The Jackson Labs, Taconic Farms, and Charles River Labs), from mice sourced from 13 barriers, with biological sex evenly distributed. For these experiments, we used an aggressive antibiotic selection strategy for our *Kp* strain, which is resistant to kanamycin and rifampicin. The unidentified gut bacterium was the only non-*Kp* bacteria that grew on selective media that we observed in any experiment and was only observed in a single vendor facility and two barriers (Taconic Farms, Germantown facility, barriers GT-20 and GT-1503, only Taconic Farms mice are shown in S1A Fig). This bacterium was only observed in three samples, all from male mice, and exhibited identical colony morphology. We selected three representative single colonies from these strains of this bacterium at random, which we named 145.1, 191.1, and 193.1 based on the mouse of origin, and identified them as *Pa* based on colony morphology and matrix-assisted laser desorption ionization–time of flight (MALDI-TOF) mass spectrometry. Whole-genome sequencing of these strains revealed that strain 145.1 was ST175 whereas 191.1 and 193.1 were unknown MLSTs, but matched 6/7 and 5/7 alleles to ST175 (as well as ST159, ST171, ST2798, ST2868, ST3748), respectively. These strains clustered with other ST175 *Pa* isolates, which grouped with Clade A strains (PAO1 major group, S1B Fig) from a previous study [43].

To determine if these wild *Pa* were able to restrict *Kp* growth, potentially explaining the observed reduction in *Kp* gut colonization density, we performed simple co-culture assays in Luria-Bertani (LB) broth. The growth of *Kp* strain KPPR1 was greatly restricted by *Pa* 145.1, 191.1, and 193.1 (Fig 1A). Given that the behavior of *Pa* is highly strain-dependent, we aimed to contextualize our finding using two well-characterized lab *Pa* strains, PAO1 and PA14. PAO1 exhibited comparatively less restriction despite its similarity of the wild *Pa* strains (Fig 1B), whereas PA14 was highly restrictive (Fig 1C). Next, we sought to determine if this phenotype was contact-dependent. To this end, we grew KPPR1 in filter-sterilized spent media of each *Pa* strain, using KPPR1 self-spent media as a control. As expected, KPPR1 growth

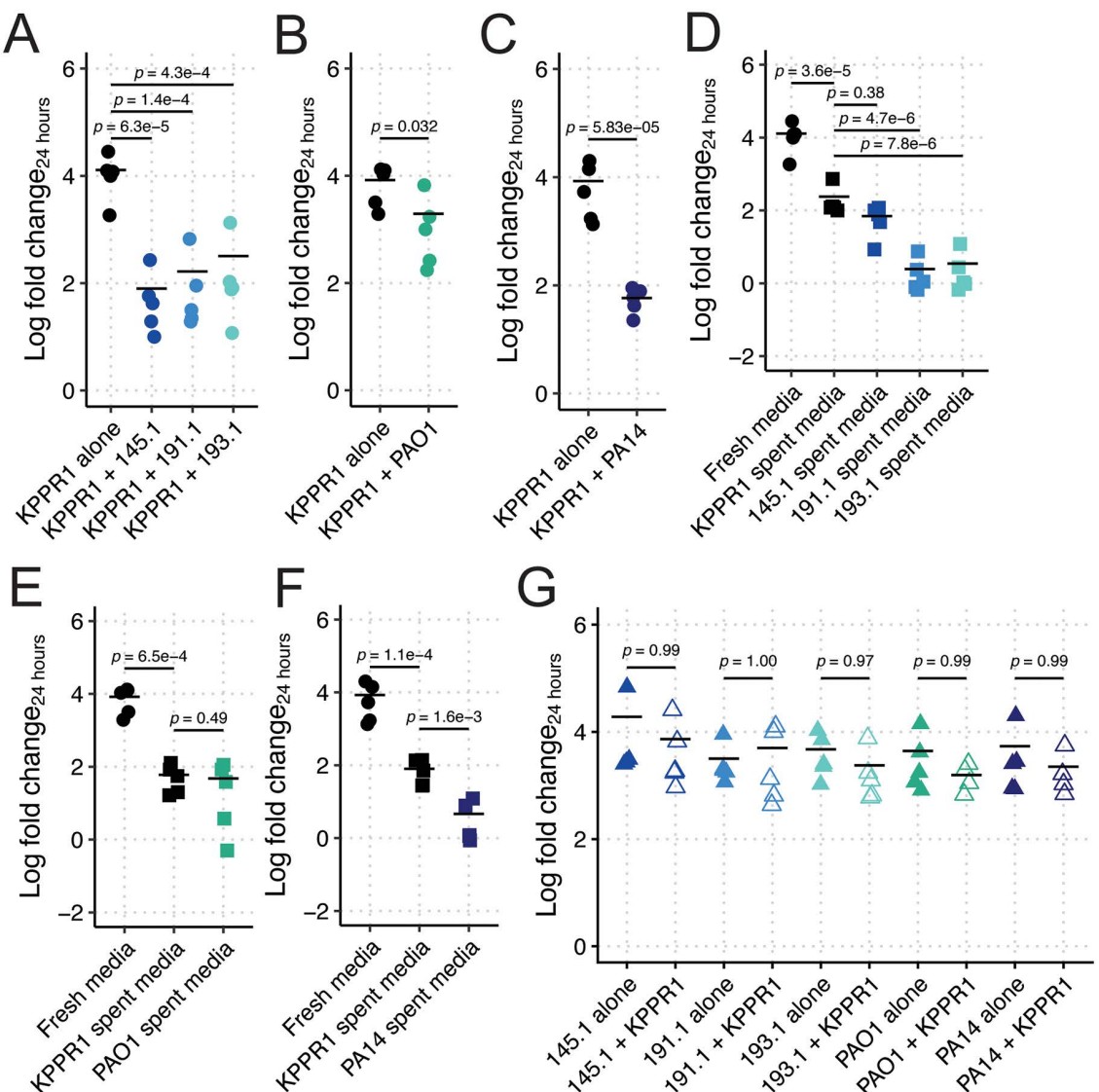

**Fig 1. *Pa* restricts *Kp* growth in LB in a strain-dependent, contact-independent manner.** KPPR1 was grown alone or in co-culture in LB with mouse-derived wild *Pa* (**A**), PAO1 (**B**), PA14 (**C**) or in filter sterilized spent media of KPPR1 or each *Pa* strain (**D–F**). *Pa* CFU was also quantified from mono- and co-cultures (**G**). For **A–G**, "Log fold change$_{24 \text{ hours}}$" = log$_{10}$(output CFU at 24 hours/input CFU). *p*-values represent Tukey multiple comparison correction following one-way ANOVA. Each data point is a biological replicate, and horizontal lines indicate the mean of each dataset. The data underlying this Figure can be found in S1 Data.

was restricted in its own sterile spent media (Fig 1D–1F); however, growth was significantly more restricted in *Pa* 191.1-, and 193.1-spent LB broth compared to self-spent media (Fig 1D). To contextualize these findings with PAO1 and PA14, we repeated these assays using these strains. Similar to co-culture results, PAO1-spent media did not significantly restrict KPPR1 growth (Fig 1E), but PA14-spent media was highly restrictive (Fig 1F). Co-culture with KPPR1 had no significant effect on *Pa* growth (Fig 1G). Collectively, these data indicate that *Pa* restriction of *Kp* growth is contact-independent, but strain-dependent.

Next, we aimed to assess the specificity of the *Kp* growth restriction phenotype to *Pa*. First, we performed co-culture competitions with the *Escherichia coli* (*Ec*) strain MG1655. Notably, *Ec* and *Kp* have similar nutritional niches, which has been exploited to reduce *Kp* gut colonization in model colonization systems [44,45]. *Ec* MG1655 was unable to restrict KPPR1 (S2A Fig). Then, we tested our approach using the *Kp* strain 13F11, which is a KPPR1 mutant with an intergenic mariner transposon insertion. We reasoned that this strain should not restrict KPPR1 growth as they are nearly identical strains. As expected, *Kp* 13F11 did not significantly restrict KPPR1 growth (S2B Fig). Neither *Ec* MG1655 nor *Kp* 13F11 spent media significantly restricted growth of KPPR1 beyond self-spent media (S2C and S2D Fig). Next, we assayed the dependency of this phenotype on growth medium. To this end, we repeated our competition assays in M9 minimal medium supplemented with 1.0% casamino acids ("M9-cas"). *Pa* restriction of KPPR1 growth in M9-cas was comparable to LB broth (S3 Fig), and *Ec* MG1655 and *Kp* 13F11 were similarly unable to restrict KPPR1 growth (S4 Fig). These data indicate that growth restriction of KPPR1 was limited to *Pa* and was insensitive to the media we tested.

## Nutrient depletion alone cannot explain *Pseudomonas* restriction of *Klebsiella*

Given that M9-cas and LB broth have similar available carbon sources, we hypothesized that the wild *Pa* strains and PA14 may be exhausting specific nutrients from these media more efficiently than that of *Ec* MG1655, *Kp* 13F11, or *Pa* PAO1. First, we measured the growth of these strains in both media types. KPPR1 had an early growth advantage over all *Pa* strains in LB broth that was mitigated by the end of the experiment, except for PA14, which reached a slightly lower final density than KPPR1 (S5A and S5B Fig). *Ec* MG1655 demonstrated a slight growth deficit compared to KPPR1, and as expected, *Kp* 13F11 growth was identical to KPPR1 (S5C and S5D Fig). *Pa* growth results in M9-cas were different than that of LB broth, wherein *Pa* grew to higher densities than KPPR1 by 24 hours (S5F and S5G Fig). *Ec* MG1655 and *Kp* 13F11 growth was identical to KPPR1 in M9-cas (S5H and S5I Fig). Given that *Pa* restriction of *Kp* was observed in both media, we concluded that the growth restriction phenotype was independent of growth characteristics, as *Pa* outgrew KPPR1 in M9-cas, but not LB broth.

Next, we aimed to determine if the metabolic potential of *Pa* and KPPR1 was similar. To this end, we assayed all five *Pa* strains, *Ec* MG1655, and KPPR1 for their ability to utilize different carbon sources using the BioLog PM1 and PM2 plates. No single carbon source was utilized more efficiently by the inhibitory *Pa* strains (PA14, 145.1, 191.1, 193.1) than the non-inhibitory strain PAO1 or KPPR1 (S6A Fig). Additionally, the *Pa* strains demonstrated a distinct carbon utilization profile compared to KPPR1 (S6B Fig), suggesting that differences in carbon source utilization were not explanatory for the restriction phenotype. Results with non-inhibitory *Ec* MG1655 were similar to the inhibitory *Pa* strains (S6 Fig), further indicating that differences in carbon source utilization did not explain *Kp* growth restriction.

Despite having distinct carbon utilization profiles, we did not exclude the possibility that *Pa* is exhausting specific nutrients from both LB broth and M9-cas. Thus, we reasoned that nutrient supplementation could fully restore KPPR1 growth. We tested the ability of KPPR1 to grow in spent LB broth supplemented with casamino acids using our single-strain spent media growth assay. We observed that growth was fully restored when self-spent LB broth was supplemented with casamino acids (Fig 2A). Interestingly, casamino acid supplementation was insufficient to restore KPPR1 growth to fresh LB broth levels in 145.1-, 191.1-, and 193.1-spent LB broth, or to the degree that supplementation restored growth in self-spent LB broth (Fig 2A). Similar results were observed for PAO1 and PA14; however, casamino acid supplementation not

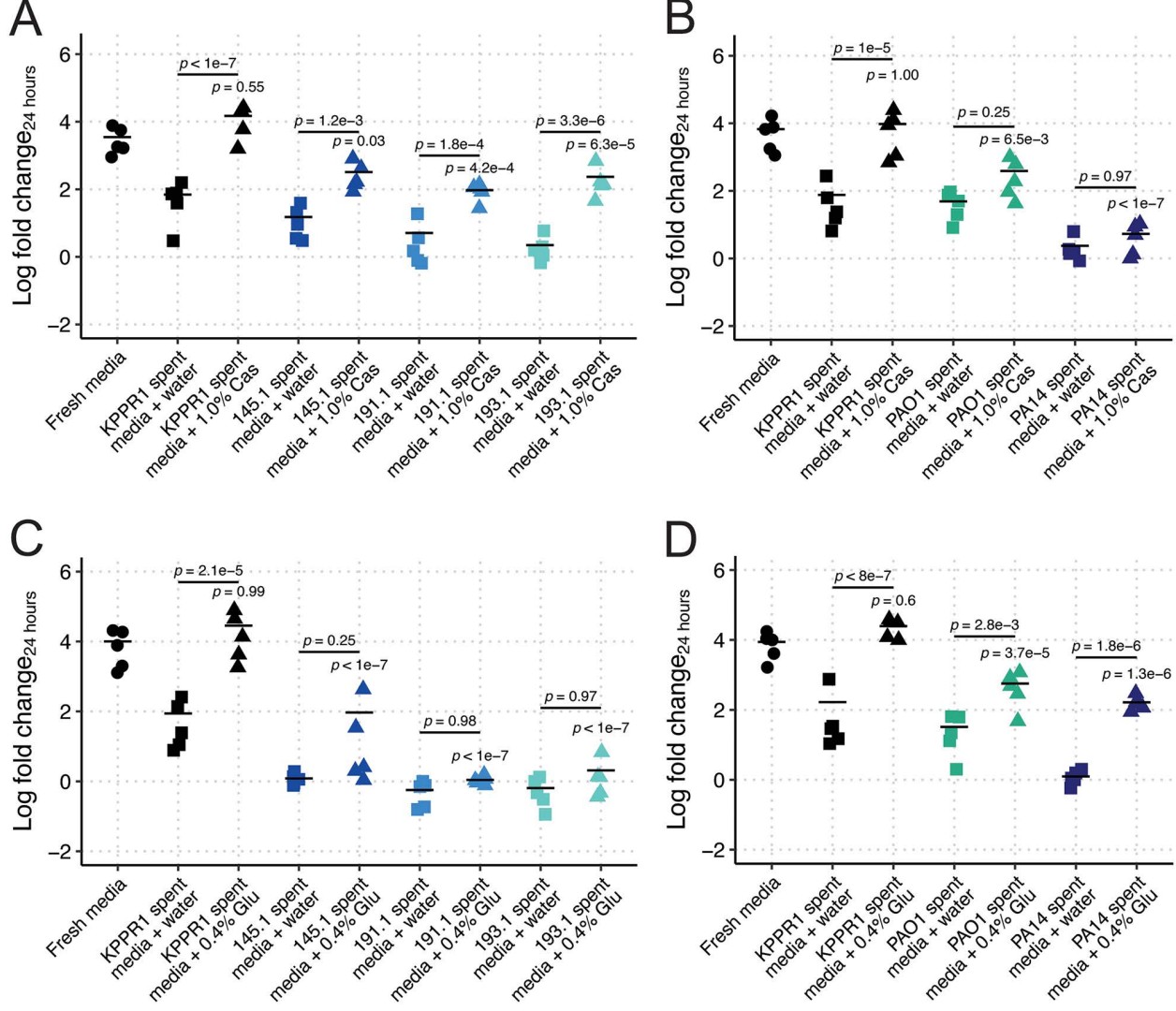

**Fig 2. Casamino acid and glucose supplementation are insufficient to complement *Kp* growth restriction in *Pa*-spent LB.** KPPR1 was grown alone or in filter-sterilized spent LB media of KPPR1 or mouse-derived wild *Pa* (**A, C**) or PAO1 and PA14 (**B, D**) supplemented with water, 1% casamino acids ("Cas," **A–B**) or 0.4% glucose ("Glu," **C–D**). For **A–D**, "Log fold change$_{24\ hours}$" = log$_{10}$(output KPPR1 CFU at 24 hours/input KPPR1 CFU). *p*-values represent Tukey multiple comparison correction following one-way ANOVA. *p*-values over columns indicate comparison to "Fresh media" condition. Each data point is a biological replicate, and horizontal lines indicate the mean of each dataset. The data underlying this Figure can be found in S1 Data.

only failed to restore growth in either PAO1 or PA14 spent LB broth but also failed to significantly enhance KPPR1 growth beyond PAO1 or PA14 spent LB broth (Fig 2B).

Finally, we tested if we were able to completely restore KPPR1 growth in *Pa*-spent LB broth by supplementing a standard carbon and energy source: glucose. Supplementation of glucose to self-spent LB broth fully restored KPPR1 growth (Fig 2C). Like casamino acid supplementation, glucose supplementation of *Pa* 145.1-, 191.1-, and 193.1-spent LB broth was insufficient to restore *Kp* growth relative to fresh LB or glucose-supplemented self-spent LB broth (Fig 2C). Glucose supplementation was more restorative of *Kp* growth in PAO1 and PA14 than casamino acids; however, KPPR1 growth

PLOS Biology

was still not restored to that of fresh media levels (Fig 2D). Collectively, these data suggest that an overlapping nutritional niche between *Kp* and *Pa* does not fully explain the restriction of *Kp* growth.

## Phenazines are necessary and sufficient for *Pseudomonas* restriction of *Klebsiella* under defined redox conditions

Given that *Pa* inhibition of KPPR1 growth is contact-independent and is at least partially independent of the nutritional niche, we reasoned that there is a factor or set of factors that *Pa* secretes to inhibit *Kp* growth. To identify candidate factors, we screened the PA14NR transposon library. To this end, we generated a GFP-expressing KPPR1 strain using the pJL1-sfGFP plasmid and screened 5,708 *Pa* transposon mutants for their ability to restrict this strain, using fluorescent intensity as a proxy for KPPR1 growth (S7A Fig and S3 Data). This screen was repeated twice, and PA14 transposon mutants from co-cultures that had a fluorescent intensity |z-score|>2.5 in both replicates were selected for validation. This yielded 18 candidate factors, 16 of which were unable to fully restrict KPPR1 growth and two that highly restricted KPPR1 growth.

The 18 transposon mutants identified as candidate factors involved in KPPR1 growth restriction were then validated for their ability to restrict KPPR1 growth. First, we determined if any of the non-restrictive phenotypes we observed were due to growth defects. Only one mutant, *tpiA* (PA14_62830), demonstrated a baseline growth defect, although this defect was modest (S7B and S7C Fig). Next, we tested the 18 mutants for their ability to restrict KPPR1 growth in co-culture and spent media assays (S7D and S7E Fig). 7/18 mutants displayed a phenotype analogous to our original screen in either the co-culture or spent media culture. Thus, this screen yielded 7 genes involved in *Pa* restriction of *Kp* growth (Table 1).

We observed that several mutants in our screen appeared to have modified phenazine production. Production of blue PYO and red PYR leads to coloration of spent culture media when in high enough concentrations. The *psqA*, *rhlR*, and *tpiA* mutants did not produce high amounts of pyocyanin (PYO, S7F Fig). As PYO has reported antimicrobial activity [46], we aimed to test their role in *Kp* restriction.

First, we confirmed our screen results using marker-less mutants to ensure that we were not observing unexpected effects of transposon mutagenesis. RhlR binds the autoinducer *N*-butanoyl-L-homoserine lactone, which is produced by RhlI, leading to expression of many genes, including the phenazine biosynthesis loci [47,48]. Both the marker-less Δ*rhlR* and Δ*rhlI* mutants were unable to restrict KPPR1 growth in our co-culture and spent media assays (Fig 3A and 3B), in both LB broth (Fig 3A and 3B), and M9-cas (S8A and S8B Fig) compared to their parental strain, PA14. Consistent with WT PA14 results (Fig 2B and 2D), supplementation of Δ*rhlR* and Δ*rhlI* spent media with casamino acids was insufficient to restore KPPR1 growth (S8C Fig), whereas glucose supplementation fully restored KPPR1 growth (S8D Fig). Importantly, the Δ*rhlR* and Δ*rhlI* mutant strains grow equivalently to PA14 in both LB and M9-cas (S5E and S5J Fig); thus, this phenotype is dependent on products of the RhlRI regulon, which includes phenazines.

**Table 1. Validated factors involved in KPPR1 growth restriction.**

| PA14 gene locus | PAO1 gene locus | Gene name | Gene description | z-score |
|---|---|---|---|---|
| PA14_01960 | PAO157 | | Putative RND efflux membrane fusion protein precursor | 4.03 |
| PA14_06570 | PA0504 | *bioD* | Dethiobiotin synthase | 6.52 |
| PA14_18520 | PA3543 | *algK* | Alginate biosynthetic protein AlgK precursor | 3.03 |
| PA14_19120 | PA3477 | *rhlR* | Acylhomoserine lactone-dependent transcriptional regulator | 4.14 |
| PA14_32750 | PA2465 | | Hypothetical protein | 3.3 |
| PA14_51430 | PA0996 | *pqsA* | Probable coenzyme A ligase | 2.9 |
| PA14_62830 | PA4748 | *tpiA* | Triosephosphate isomerase | 3.05 |

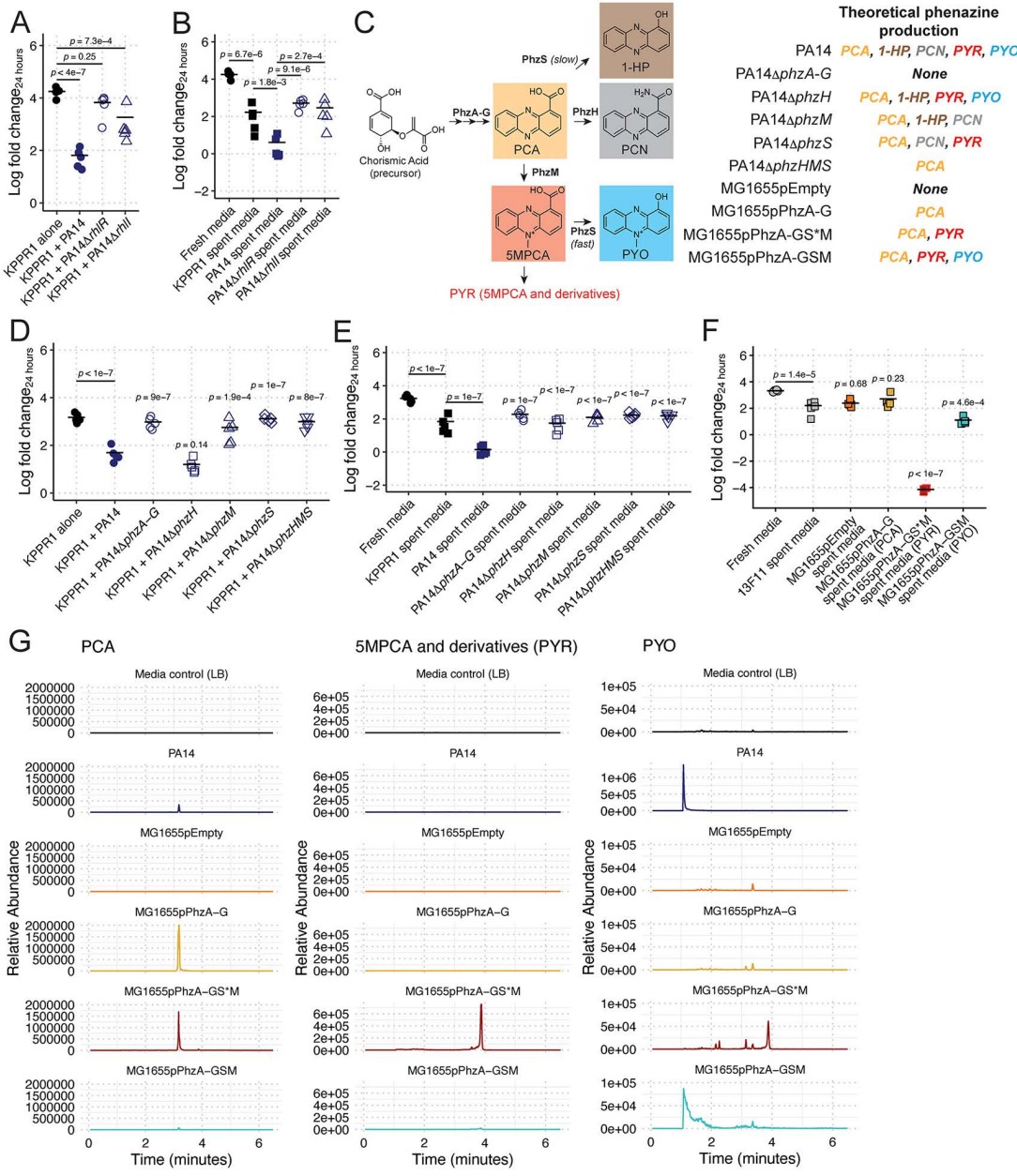

**Fig 3. Phenazines are necessary and sufficient for _Kp_ growth restriction.** KPPR1 was grown alone or in co-culture in LB with WT PA14, PA14Δ_rhlR_, or PA14Δ_rhlI_ (**A**) or in filter-sterilized spent media of KPPR1 or each _Pa_ strain (**B**). Phenazine biosynthesis (**C**) is under the transcriptional control of RhlIR. KPPR1 was grown alone or in co-culture in LB with WT PA14, PA14Δ_phzA-G_, PA14Δ_phzH_, PA14Δ_phzM_, PA14Δ_phzS_, or PA14Δ_phzHMS_ (**D**) or in filter-sterilized spent media of KPPR1 or each _Pa_ strain (**E**). Additionally, 13F11 (Kan^R KPPR1 variant) was grown in filter-sterilized spent media of 13F11 or MG1655 containing an empty vector (pEmpty) or constitutively expressing PCA (pPhzA-G), PYR (pPhzA-GS*M), and PYO (pPhzA-GSM), **F**). Datapoints outlined in red are below the limit of detection (200 CFU/mL). For **A–B** and **D–F**, "Log fold change$_{24 \text{ hours}}$" = log$_{10}$(output KPPR1 CFU at 24 hours/input KPPR1 CFU). Each data point is a biological replicate, and horizontal lines indicate the mean of each dataset. _p_-values represent Tukey multiple comparison correction following one-way ANOVA. For **D**, **E**, and **F**, _p_-values above individual columns indicate comparison to "KPPR1 + PA14," "PA14 spent media," and "13F11 spent media" groups, respectively. LC-MS was used to quantify phenazine secretion from _Ec_ and _Pa_ strains (**G**). Note that the scale for the PA14 PYO chromograph is larger than the other PYO chromographs to accommodate the large peak. PA14 and MG1655 pPhzA-GSM chromographs on the same scale can be found in S15B Fig. The data underlying this Figure can be found in S1 Data.

Then, we repeated our co-culture and spent media assays with strains containing deletions in the *phzA-G1-2*, *phzH*, *phzM*, *phzS*, and *phzHMS* loci. PhzA-G are necessary to synthesize phenazine-1-carboxylic acid (PCA), which is converted to 5MPCA by PhzM and from 5MPCA to PYO by PhzS or directly converted from PCA to 1-hydroxyphenazine (1-HP) by PhzS [49]. Additionally, PhzH converts PCA to phenazine-1-carboxamide (PCN). Pyorubin (PYR), which collectively refers to 5-methylphenazine-1-carboxylate (5MPCA) and its spontaneously synthesized derivatives including 5-methyl-7-amino-1-carboxyphenazinium betaine (Aeruginosin A) and 5-methyl-7-amino-1-carboxy-3-sulfophenazinium betaine (Aeruginosin B) [50,51], is an intermediate metabolite in the PYO biosynthesis pathway that has previously been reported to have antimicrobial activities [52]. *Pa* encodes two *phzA-G* loci, and deletion of both loci renders *Pa* unable to synthesize PCA, PYO, 5MPCA (and thus, PYR), 1-HP, or PCN. Deletion of *phzH* ablates PCN production, deletion of *phzS* ablates PYO and 1-HP production, deletion of *phzM* ablates 5MPCA and PYO, and deletion of *phzHMS* ablates production of PYO, PYR, 1-HP, and PCN but maintains PCA production (Fig 3C). Interestingly, we observed significantly less *Kp* growth restriction by the Δ*phzA-G*, Δ*phzM*, Δ*phzS*, and Δ*phzHMS* mutants in co-culture, and no impact with the Δ*phzH* mutant (Fig 3D). Similar results were observed in our spent media assay, though the Δ*phzH* mutant displayed an entirely contact-dependent growth restriction of Kp which was not observed in other mutants, suggesting PYO and PYR were not responsible for the activity of this mutant (Fig 3E). Of note, we did not identify any *phz* mutant in our initial transposon screen. Given that one *phzA-G* locus is sufficient for phenazine production, we would not expect to identify these mutants as candidates. In fact, *phzA1* (PA14_09480) was a candidate that inhibited *Kp* more than average, suggesting that there may be overexpression of phenazines; however, this observation failed to validate following the screen (S7D and S7E Fig). Additionally, the known phenazine biosynthesis regulator PqsE did not meet our significance criteria, but the data fit the general trend of a less inhibitory strain (trial 1 *z*-score = 2.13, trial 2 *z*-score = 1.19). We were surprised that the *phzM* or *phzS* mutants were not identified in our transposon screen, as these mutants should have disrupted phenazine biosynthesis; however, these transposon interruptions may not sufficiently inactivate gene function, whereas clean knockouts yielded higher-confidence data (Fig 3D). Collectively, this suggests that the phenazine PYO and/or PYR are necessary for the observed growth restriction phenotype.

Most studies investigating the antimicrobial mechanisms of phenazines have focused on PYO. The prevailing model for PYO's antimicrobial activity involves the induction of oxidative stress: PYO readily diffuses across cellular membranes and promotes the formation of superoxide radicals, leading to cellular toxicity (reviewed in [53]). Thus, to determine if PYO is sufficient for restriction of *Kp* growth, we used a heterologous expression system in *Ec* MG1655 to constitutively produce PCA, PYR, and PYO, testing *Kp* (strain 13F11, Kan^R KPPR1 strain) in the resulting spent media. As expected, the empty vector and PCA over-producing strains failed to inhibit *Kp* growth beyond that of self-spent media (Fig 3F). Conversely, the spent media from the PYO over-producing strain inhibited *Kp* growth (Fig 3F) to a similar degree as WT PA14 (Fig 1F), demonstrating that PYO is sufficient to restrict *Kp* growth.

### Pyorubin exhibits bactericidal activity

Surprisingly, the spent media from the PYR over-producing strain was completely bactericidal (Fig 3F). Of note, PYR over-production was a result of a spontaneous frameshift mutation at T998 of *phzS* (annotated as "pPhzA-GS*M") that ablated the final step of PYO production, leading to rapid accumulation 5MPCA and its derivates (PYR). High-resolution mass spectrometry of the spent medium of this strain confirmed the presence of 5MPCA and that it was absent in the spent media of other heterologous expression strains (Fig 3G). A previous report of a 5MPCA-sensitive microbe indicated that the activity of 5MPCA is dependent on its oxidation state, wherein its reduced form is more active than its oxidized form [52]. To determine if this is true for the activity of PYR against *Kp*, we titrated a reducing agent into spent media from our PYR-producing *Ec* strain and measured its ability to kill *Kp*. Consistent with previous reports, the addition of a reducing agent (generating reduced 5MPCA) increased anti-*Kp* activity (S9 Fig). Given that mutation of PhzS led to PYR over-expression in our *Ec* system, we were curious as to why our PA14 *phzS* mutant was not equally inhibitory (Fig 3D–3F),

as this mutant has the genes necessary to produce PYR. We performed high-resolution mass spectrometry of the spent media from this mutant, as well as a *phzS* transposon mutant from our PA14 transposon library screen. We did not detect substantial PYR levels from these strains (S10A Fig), suggesting that at least partially intact PhzS is required to produce PYR or that PA14 has alternative pathways to modify PYR. Finally, we did not detect 1-HP secretion from any of our *Pa* or *Ec* strains (S10B Fig). Taken together, these data show that specific phenazines, namely PYO and PYR, are both necessary and sufficient to restrict *Kp* growth.

Next, we aimed to assess the antimicrobial activity of phenazines against KPPR1. First, we measured the minimum inhibitory (MIC) and minimum bactericidal concentrations (MBC) of pure PYO, as it was unclear from our co-culture approach if this compound was bacteriostatic or bactericidal. To replicate our co-culture and spent media culture assays, we measured the MIC and MBC of PYO. We performed these assays in both fresh LB and PA14Δ*phzA-G* spent LB, the latter of which we used to mimic the nutrient conditions in which *Kp* is experiencing phenazine-induced stress, as nutrient availability will modulate phenazine sensitivity by changing the cellular redox state [54]. The MIC of PYO was 8- and 32-fold lower than the MBC (Fig 4A and 4B), in fresh and spent media, respectively, indicating that PYO is primarily bacteriostatic, which is consistent with the ability of PYO to arrest bacterial respiration [46], as opposed to kanamycin, which has an MIC only 2-fold lower than the MBC, indicating bactericidal activity (Fig 4A and 4B). Then, we sought to confirm our co-culture data that demonstrated that PCA and 1-HP had no activity against KPPR1 (Fig 3D). The MICs of both compounds in pure form were higher than that of PYO in both fresh (Fig 4C) and spent (Fig 4D) media. Of note, the calculated PCA MIC is likely an overestimation of activity, as the large amount of DMSO (>10% volume at ≥250 μg/mL PCA) in culture medium used to maintain PCA solubility likely affected KPPR1 growth. Nonetheless, these data support our findings that PYO restricts *Kp* growth, whereas PCA and 1-HP have little or no effect.

We determined that PYR is primarily bactericidal, as the MIC is at most 2-fold lower (Fig 4E) than the concentration where bactericidal activity was detected (Fig 3F). Interestingly, PYO and PYR display concentration-dependent additive (0.5<Fractional Inhibitory Concentration<4, % MG1655pPhzA-GS*M spent media=3.13%) and synergistic effects (Fractional Inhibitory Concentration>0.5, % MG1655pPhzA-GS*M spent media=6.25%, 12.5%, 25%, Fig 4F).

Finally, we estimated the amount of PYO and PYR for each of our WT *Pa* strains produced in conditions identical to our spent media assay. PYO results were concordant with our co-culture and spent media assays, wherein PAO1 produced the least PYO (0.272 μg/mL [1.29 μM]) and was the least restrictive strain, followed by 145.1 (4.02 μg/mL [19.1 μM]), 191.1 (15.5 μg/mL [73.7 μM]), 193.1 (17.4 μg/mL [82.8 μM]), and PA14 (20.7 μg/mL [98.5 μM], Fig 4G). All five strains displayed little PYR production (Fig 4H), measured at $Abs_{500}$ (S11A Fig), concordant with high-resolution mass spectrometry for PA14, which displayed no 5MPCA production (Fig 3G). Only 193.1 displayed an $Abs_{500}$ higher than PA14, suggesting limits to the analytical sensitivity of this assay. Nonetheless, these data demonstrate that phenazine production by the five *Pa* strains is explanatory for our spent media assay results, wherein the three inhibitory strains (191.1, 193.1, and PA14) produce phenazines at or near the MIC, and the two less-inhibitory strains (PAO1 and 145.1) produce phenazines well below the MIC.

## Environmental oxygen tunes phenazine potency

Some studies have shown that PYO remains active under anoxic conditions [55–57]. These findings suggest that PYO can remain chemically redox-active in the absence of oxygen; however, antimicrobial effects may still be oxygen-linked if they require oxygen-mediated redox cycling/ROS generation or maintenance of PYO in its oxidized state. To determine if this is the case for KPPR1, we measured the MIC of PCA, 1-HP, and PYO under anaerobic conditions. We observed a significant reduction of PYO activity under anaerobiosis in both fresh (Fig 4C) and spent (Fig 4D) media. Additionally, we repeated our co-culture competition assays under anaerobic conditions. As with our MIC studies, we did not observe *Pa* growth restriction of KPPR1 (S12A Fig). We also repeated our spent media growth assays using the heterologous phenazine expression system. Like our co-culture results, we did not observe any growth restriction or bactericidal activity in the case of PYR under anaerobic conditions (S12B Fig).

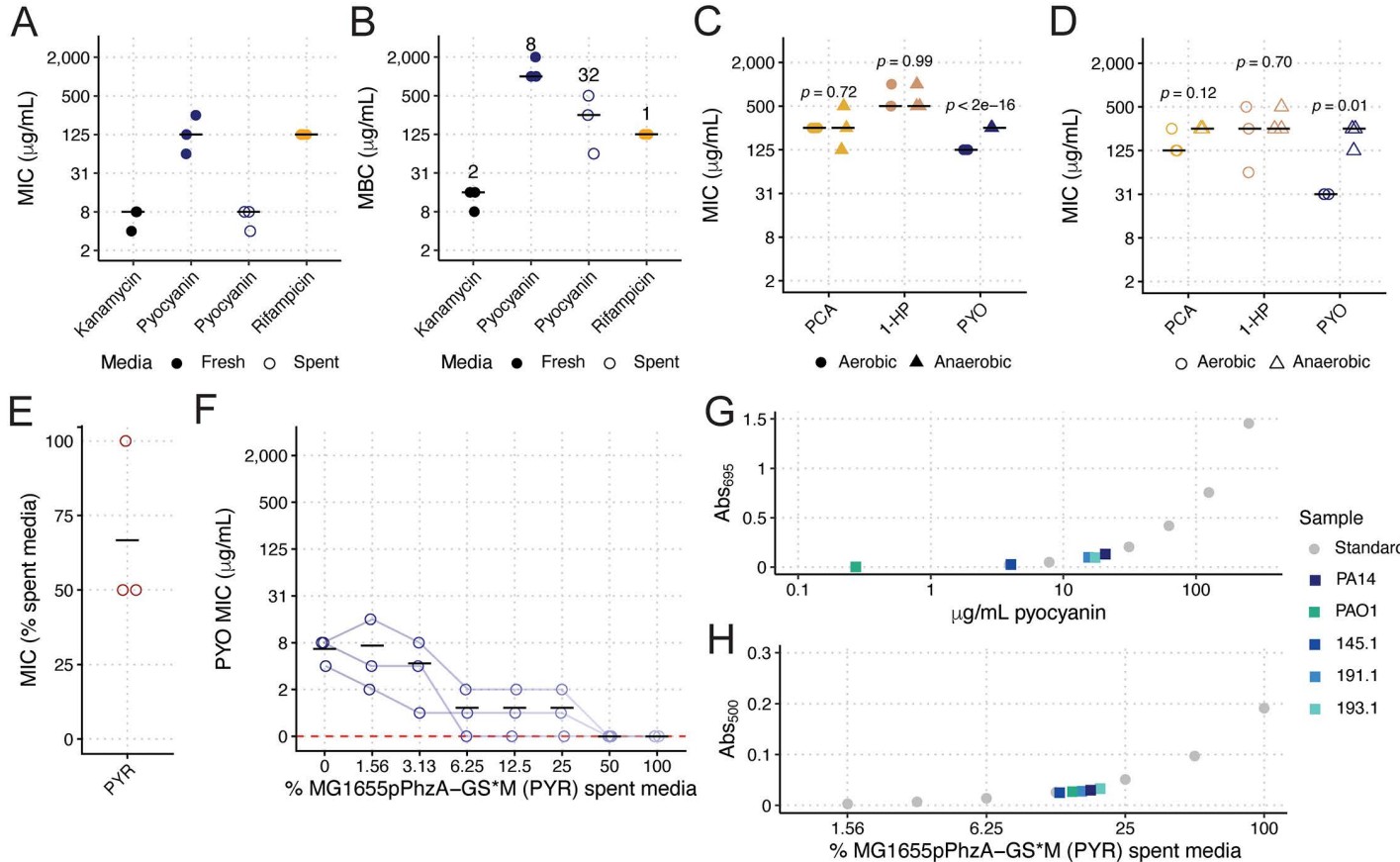

**Fig 4. Minimum inhibitory and bactericidal concentrations of phenazines.** The minimum inhibitory concentration (MIC) and minimum bactericidal concentrations (MBC) were determined for KPPR1 for kanamycin, to which KPPR1 is sensitive, pure PYO, and rifampicin, to which KPPR1 is resistant (**A, B**). The numbers above each column in **B** represent the median MBC:MIC ratio. The MIC of pure PCA, 1-HP, and PYO was measured in aerobic and anaerobic conditions in fresh (**C**) and spent (**D**) media. The MIC of PYR (MG1655pPhzA-GS*M spent media) was also calculated (**E**). The effect of PYR concentration on the PYO MIC was also measured (**F**). The $Abs_{695}$ and $Abs_{500}$ of filter-sterilized spent LB from mouse-derived wild Pa, PAO1, and PA14 were measured and PYO concentrations were interpolated from a PYO (**G**) and PYR (**H**) standard curve. Each data point is a biological replicate, and horizontal lines indicate the mean of each dataset. For **C, D**, $p$-values represent Tukey multiple comparison correction following one-way ANOVA, comparing aerobic vs. anaerobic MICs within each phenazine. For **G, H**, only median values of three biological replicates are shown. The data underlying this Figure can be found in S1 Data.

Finally, to determine if phenazine activity is dependent on respiration, we tested the MIC of PYO in conditions that favor anaerobic respiration. To this end, we repeated our MIC experiments under anaerobic conditions but grew *Kp* in M9 minimal medium with 0.5% glucose ("M9-glu"), conditions in which *Kp* predominantly undergoes fermentation, and M9-glu supplemented with 10 mM $NaNO_3$, which permits anaerobic respiration by nitrate acting as an alternative electron acceptor [58,59]. Addition of nitrate had no effect on the PYO MIC (S12C Fig), indicating that nitrate-permitting anaerobic respiration does not restore PYO activity. Thus, in our system, PYO antimicrobial action may require oxygen-dependent redox cycling or maintenance of oxidized PYO, supporting dependency on environmental oxygen.

## Species-level susceptibility landscape reveals consistent sensitivity

Next, we aimed to determine the translatability of these findings to a diverse set of human *Pa* and *Kp* clinical isolates derived from a variety of infection sources. First, we screened the ability of 198 clinical *Kp* isolates to grow in PA14

spent media. The growth of all 198 strains was highly restricted, like KPPR1 (Fig 5A), indicating a universal phenotype. Co-culture assays validated these findings (S13 Fig). We then screened a subset of these clinical *Kp* isolates (*N* = 41) for growth in the spent media of our PCA, 5MPCA, and PYO constitutively expressed *Ec* strains. Of note, to accommodate the growth of our clinical *Kp* isolates, phenazine-containing spent media was generated in the absence of antibiotic

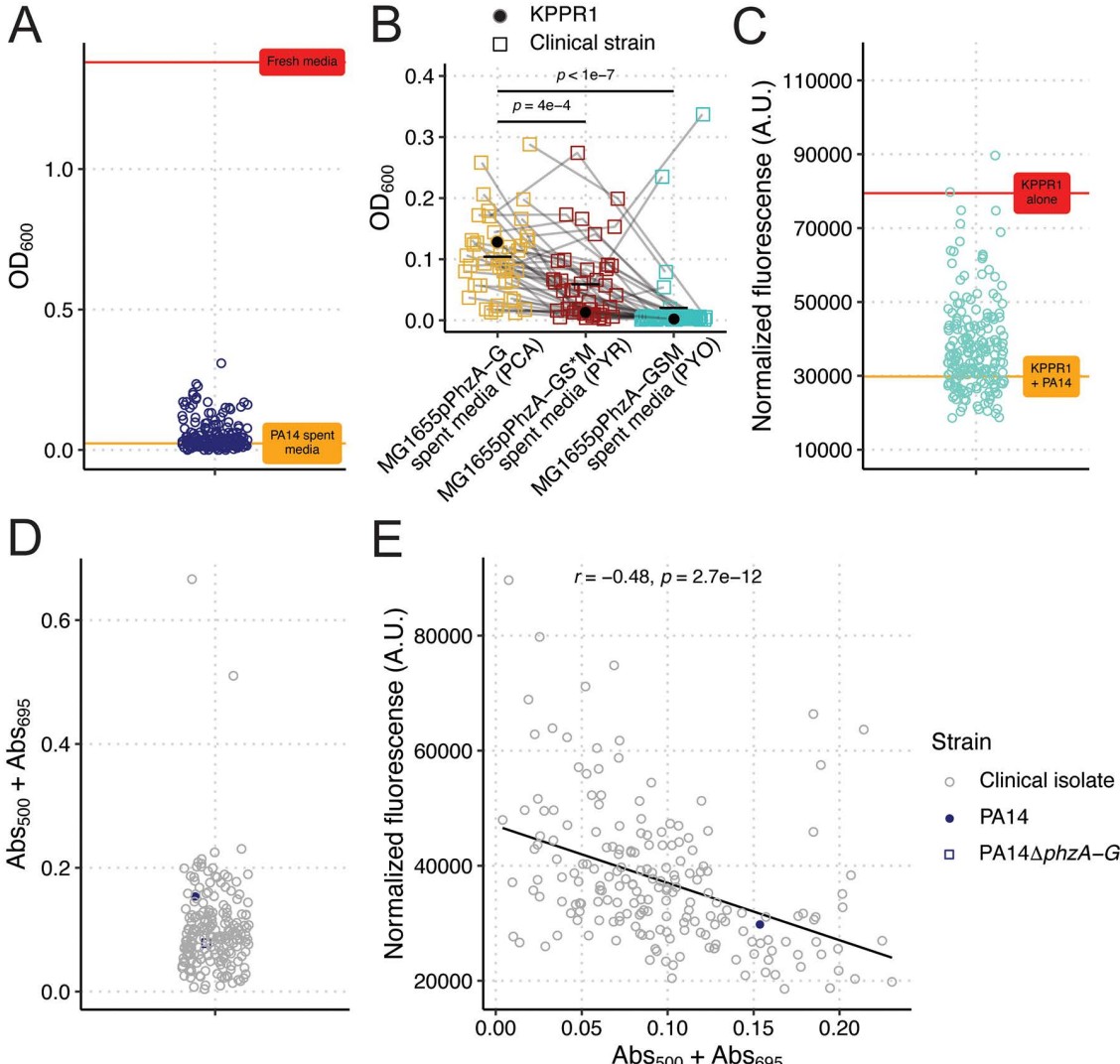

**Fig 5. Phenazines universally restrict *Kp* growth, but restriction is strain-specific.** *Kp* clinical isolates (*N* = 198) were grown in sterilized spent PA14 LB (**A**). $OD_{600}$ was measured at 24 hours. For **A**, each data point represents the median read of each culture. The red line represents mean $OD_{600}$ of KPPR1 in fresh LB, and the orange line represents the mean $OD_{600}$ of KPPR1 in PA14 spent LB. A subset of clinical *Kp* isolates (*N* = 41) and KPPR1 were grown in filter-sterilized spent media of MG1655 constitutively expressing PCA (pPhzA-G), PYR (pPhzA-GS*M), and PYO (pPhzA-GSM, **B**). $OD_{600}$ was measured at 24 hours. For **B**, each data point represents the median read of each culture, and gray lines connect identical isolates. *p*-values represent Tukey multiple comparison correction following one-way ANOVA. *Pa* clinical isolates (*N* = 194) were co-cultured in LB with GFP-expressing KPPR1 (**C**). Fluorescence (arbitrary units) was measured at 24 hours and normalized to culture density ($Abs_{600}$). For **C**, each data point (blue) represents the median of each co-culture. The red line represents the mean normalized fluorescence for KPPR1, and the orange line represents the mean normalized fluorescence for KPPR1 + PA14. For *Pa* clinical isolates (*N* = 194), WT PA14, or PA14Δ*phzA-G*, phenazine production was measured in LB broth at $Abs_{500}$ (PYR) and $Abs_{695}$ (PYO, **D**) and correlated (*N* = 192) to growth restriction results from **C** (Spearman correlation test, **E**). The data underlying this Figure can be found in S1 Data.

selection to maintain the expression plasmids, and correspondingly, phenazine yield was lower than that in Fig 3F. Similar to our results with *Kp* 13F11, we observed that PYR and PYO significantly restricted the growth of our clinical *Kp* isolates (Fig 5B); however, this phenotype was strain-dependent, wherein some *Kp* strains were more resistant to PYO and PYR growth restriction than others.

Then, we screened the ability of 194 *Pa* strains to restrict KPPR1 growth. Interestingly, KPPR1 restriction was highly strain dependent (Fig 5C), akin to our findings with PAO1 and PA14 (Fig 1B and 1C). Co-culture assays validated these findings (S14 Fig). Given that phenazines are necessary for PA14 restriction of KPPR1, we then tested phenazine production in our screening conditions (S11A Fig). The clinical *Pa* strains produced phenazines to varying degrees, with many producing very little (Figs 5D, S11B, and S11D). PYO and PYR levels as measured by absorbance were significantly correlated (S11F Fig). Of note, phenazine measurements in this assay are based on absorbance (S11A Fig), thus, there may be spectral overlap between phenazines or other non-phenazine compounds produced that are detected at the wavelengths we tested. Forty-seven *Pa* strains had $Abs_{500}$ values higher than PA14, suggesting that these strains may secrete PYR (S11D Fig). The ability of clinical *Pa* strains to restrict *Kp* significantly inversely correlated with phenazine production (Figs 5E, S11C, and S11E); however, the correlation between phenazine production and *Kp* growth restriction was weak ($r = -0.48$, $r^2 = 0.23$), wherein many *Pa* strains produce little or no phenazine but are highly restrictive and others less restrictive but appear to produce high levels of phenazine. The two strains that produced high levels of phenazines in Fig 5D were excluded from the analyses in Fig 5E, as these strains produced a pigment similar in appearance to pyomelanin, resulting in high absorbance values not due to phenazine production.

To test if other secreted *Pa* effectors may be a factor in *Kp* growth restriction as our data suggested, we selected two *Pa* strains that we validated as inhibitory in our co-culture assay (S14 Fig): JV46 and JV69. In our screen, JV46 was slightly more inhibitory than PA14 (mean normalized fluorescence = 26,524.17 versus 29,785.32) and JV69 was less inhibitory than PA14 (mean normalized fluorescence = 35,396.03 versus 29,785.32), yet these strains produced similar levels of phenazines (JV46 $Abs_{500} + Abs_{695} = 0.154$, JV69 $Abs_{500} + Abs_{695} = 0.103$, PA14 $Abs_{500} + Abs_{695} = 0.154$). The spent media of JV69 was inhibitory to a comparative level as PA14, where JV46 was bactericidal (S15A Fig). Of note, the $Abs_{500}$ measurement of JV46 spent media was lower than that of PA14 (0.07 versus 0.086), suggesting that the bactericidal phenotype is not due to increased PYR secretion, which we confirmed by mass spectrometry (S15B Fig). This indicates that phenazine production is likely only one mechanism by which *Pa* restricts *Kp* growth. Collectively, we concluded that at high concentrations, *Kp* is universally sensitive, but at low concentrations, sensitivity is strain dependent.

### Phenazine-dependent restriction of *Klebsiella pneumoniae* is site-specific

There is growing interest in using live biotherapeutics to preventatively decolonize *Kp* and other gut colonizing pathogens to lower infection risk [44,45,60]; however, current approaches largely depend on manipulation of the metabolic niche, rather than leveraging direct growth inhibition. Given our isolation of the wild *Pa* strains from mouse gut samples with low *Kp* colonization and their ability to directly restrict *Kp*, we hypothesized that these *Pa* strains can exclude *Kp* from the gut. To test this hypothesis, we colonized C57Bl6/J mice with *Pa* 191.1. To open the gut niche for colonization, mice were treated with ampicillin prior to *Pa* colonization. 191.1 was able to stably colonize the gut (S16A Fig), though colonization density was low. One week following *Pa* colonization, mice were colonized with KPPR1. An antibiotic-treated group not colonized by 191.1 was included as a control. No difference in KPPR1 gut colonization density was observed between mono-colonized and co-colonized mice (S16B and S16C Fig), potentially due to low 191.1 colonization. Next, we hypothesized that the intact gut microbiome may be required to observe a restrictive phenotype, as the gut microbiome of the mice in our initial in vivo experiment was intact (S1A Fig). To this end, we used an *ex vivo* approach to test this hypothesis. Whole large intestinal contents of C57Bl6/J mice were collected and resuspended in sterile phosphate-buffered saline (PBS) to generate gut microbiota-replete competition media. KPPR1 was anaerobically competed in this media against 145.1, 191.1, and 193.1. All *Pa* strains were viable in these conditions (S17A Fig); however, no reduction in KPPR1 CFUs

was observed (S16D Fig). Together, these results indicate that the wild *Pa* strains are unable to restrict *Kp* growth in intestinal contents. Of note, this corroborates our finding that phenazine-dependent *Kp* growth restriction is dependent on environmental oxygen.

Given our gut findings that were contradictory to our original observations, we turned to other relevant body sites for interrogating the interactions between *Pa* and *Kp* where phenazine production may be important. Interestingly, stratification of clinical *Pa* growth restriction data based on the clinical site of origin suggested site specificity (Fig 6A and S1 Table). *Pa* isolated from the blood and urine were more restrictive compared to those isolated from the respiratory tract or wounds. To determine if there is a causal relationship between phenazine production and site-specific restriction of *Kp* growth, we aimed to test phenazine-dependent *Kp* growth restriction in conditions that mimic respiratory and urine infections. To this end, co-culture competitions were performed in murine bronchoalveolar lavage fluid (BALF) and *ex vivo* bladder homogenate. Both *ex vivo* tissues sustained the growth of both *Kp* and *Pa* (S17B and S17C Fig). Interestingly, we did not detect phenazine-dependent growth restriction of *Kp* in murine BALF (Fig 6B), supporting a diminished role in *Kp* lung co-infection. Conversely, we detected a significant phenazine-dependent growth restriction phenotype in *ex vivo* bladder homogenate (Fig 6C), consistent with a more important role of phenazines during UTI. Collectively, these data demonstrate that the impact of phenazines on *Pa-Kp* interaction, and therein, infection outcomes, is site-specific.

### Effects of phenazines on the growth of diverse bacteria

To assess the species-specificity of phenazine-dependent growth restriction, we repeated our heterologous expression spent media growth assays with a panel of different bacteria, specifically those that are important causes of UTIs. We included two *K. oxytoca* strains (*Ko*, JV282, JV453) to test within the *Klebsiella* genus, two *Ec* strains (UTI89, CFT073) to test outside the *Klebsiella* genus, and an *Enterococcus faecalis* strain (JV25) isolated from a mouse gut to test a Gram-positive species. The effects of PCA, PYO, and PYR on gram-negative bacteria were similar to that of *Kp*, wherein

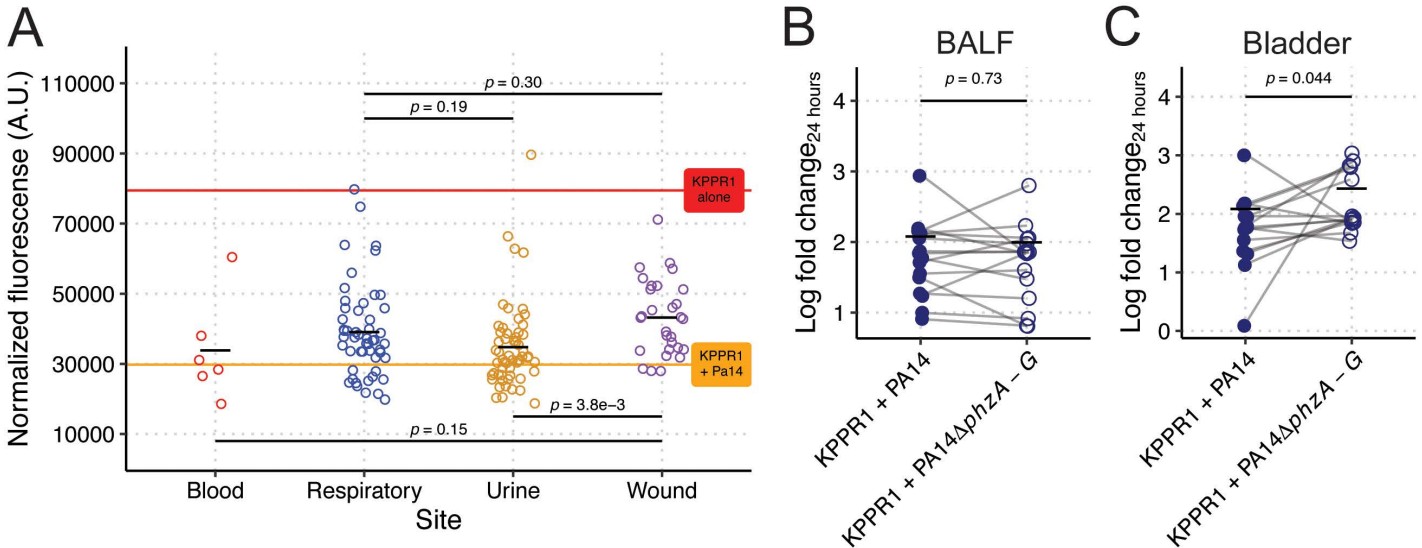

**Fig 6. Phenazine-dependent restriction of *Kp* is site-specific.** Mean normalized fluorescence for KPPR1 from co-culture with *Pa* clinical isolates (*N* = 194) in LB from Fig 5C was stratified by clinical culture site (**A**). The red line represents mean normalized fluorescence for KPPR1, and the orange line represents the mean normalized fluorescence for KPPR1 + PA14. KPPR1 was grown in co-culture in bronchoalveolar lavage fluid (**B**) or *ex vivo* bladder homogenate (**C**) with WT PA14 or PA14Δ*phzA-G*. For **B–C**, "Log fold change$_{24\text{ hours}}$" = log$_{10}$(output KPPR1 CFU at 24 hours/input KPPR1 CFU). Each data point is a biological replicate, connecting lines indicate corresponding *ex vivo* samples, and horizontal lines indicate the mean of each dataset. *p*-values represent unpaired *t* tests. The data underlying this Figure can be found in S1 Data.

they exhibited growth restriction by *Pa* PA14 and high sensitivity to PYR and PYO, with overall sensitivity varying by strain (Fig 7A–7D). The *Ec* strains and the *Ko* strain JV282 also exhibited sensitivity to PCA. Interestingly, our representative Gram-positive species, *E. faecalis*, did not exhibit a high degree of sensitivity to *Pa* PA14, PYR, or PYO, but modest sensitivity to PCA (Fig 7E). Collectively, this suggests that phenazines have greater effects on Gram-negative bacteria than Gram-positive bacteria; however, a more extensive study is required to test this premise. Nonetheless, these data indicate the effects of phenazines are not specific to *Kp* and the data shown here are likely applicable to a wide array of bacteria.

## Discussion

In this study, we investigated the ability of *Pa* to restrict *Kp* growth and demonstrated that it can do so in a contact-independent manner. These findings are partially dependent on nutrient competition; however, the synergistic and/or additive effects of *Pa*-secreted phenazines account for the majority of this growth restriction. The phenazine PYO, well known for its antibacterial activity, has bacteriostatic effects on *Kp*, while the less well-characterized PYR has bactericidal effects, at least at high concentrations. This was also observed for *Ko* and *Ec*, but not *E. faecalis*. Interestingly, these phenotypes are highly dependent on the *Pa* strain, wherein phenazine production is a strong correlate of *Kp* restriction, though again, is not completely explanatory. Rather, some *Pa* strains restrict *Kp* growth without producing high levels of phenazines. Further complicating this relationship, environmental conditions also impact the outcome of *Pa*-*Kp* competition. We observed phenazine-dependent growth restriction in conditions that mimic bladder infection but not lung infection, which corresponded to experimental interrogation of our collection of clinical isolates. Collectively, our study shows that interactions between *Pa* and *Kp* are highly complicated, and that variables like genetic composition and environmental conditions are key determinants of fitness outcomes. This provides a potential explanation for study-to-study variation and lays a foundational approach for interrogation of microbial interactions during pathogenesis wherein many variables that may impact experimental outcomes should be considered.

The toxicity of PYO to various bacterial species has been shown previously, whilst 5MPCA toxicity has been previously observed in *Candida albicans* [52,53]. It has been suggested that in *Pa*, 5MPCA is rapidly converted to PYO to prevent accumulation of a toxic intermediate [61]. The spontaneous frameshift in PhzS in our MG1655pPhzA-GS*M strain leads to increased production of PYR and increased *Kp* toxicity, supporting the hypothesis that it is more toxic than PYO. PYO has been shown to accept electrons from the electron transport chain and donate them to oxygen, disrupting cellular redox balance and generating cytotoxic reactive oxygen species [54]. Whilst PYR toxicity was dependent on aerobic conditions, its high redox potential [62] make oxygen reduction unfavorable [63]. Our findings support a general principle that environmental redox gates the potency of phenazine-mediated interference. Phenazines such as PYO undergo redox cycling that depends on molecular oxygen and the local electron-acceptor milieu; in turn, this dictates whether growth inhibition is primarily bacteriostatic or bactericidal. Prior work shows PYO perturbs intracellular redox and central carbon flux, consistent with oxygen-linked toxicity, while broader phenazine reviews underscore how small structural differences shift reactivity across niches [54,56]. Furthermore, *Ec* MG1655 produced large titers of PYR that were bactericidal to *Kp*, despite genetic and metabolic similarities between these species [44]. This suggests a specific mode of PYR toxicity other than reactive oxygen species generation. One major difference between aerobic respiration in *Ec* and *Klebsiella* is the dependence of the latter on type II NADH-dehydrogenases, especially in urinary media [64]; however, these studies were performed with *K. aerogenes*. In *Pa*, these enzymes are the predominant phenazine reductases, suggesting phenazine inhibition of *Kp* could be caused by it acting as an electron acceptor, uncoupling NADH oxidation to ATP synthesis in the electron transport chain [65]. Further work is required to determine the mechanism underlying *Kp* phenazine toxicity.

The initial observation that was the inception of this study was the isolation of the wild *Pa* strains (145.1, 191.1, 193.1) from murine large intestinal contents (S1A Fig). Yet, deeper interrogation revealed that phenazine-dependent growth restriction of *Pa* by *Kp* is oxygen-dependent (Fig 4). This result is surprising, as our original *Kp* gut colonization experiments were performed in the context of an unperturbed gut microbiome, which would maintain a largely anoxic

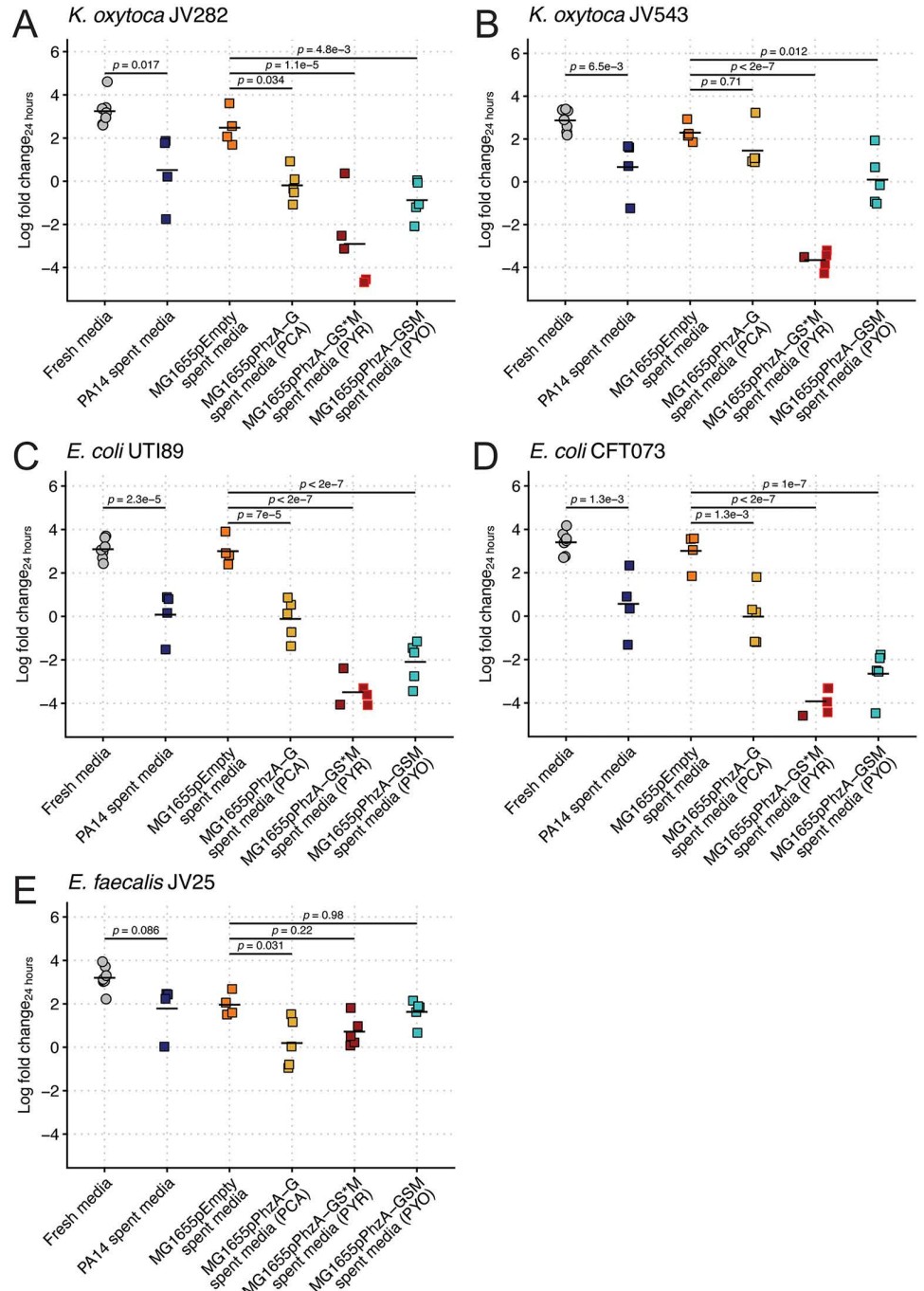

**Fig 7. Phenazine-dependent restriction of diverse bacteria.** *K. oxytoca* JV282 (**A**) and JV543 (**B**), *Ec* UTI89 (**C**) and CFT073 (**D**), and *E. faecalis* JV25 (**E**) was grown alone in fresh LB broth or in filter-sterilized PA14, MG1655pEmpty, MG1655pPhzA-G (PCA), MG1655pPhzA-GS*M (PYR), or MG1655pPhzA-GSM (PYO) spent media. For **A–E**, "Log fold change$_{24 hours}$" = log$_{10}$(output CFU at 24 hours/input CFU). *p*-values represent Tukey multiple comparison correction following one-way ANOVA. Each data point is a biological replicate, and horizontal lines indicate the mean of each dataset. Heterologous expression *Ec* strains were grown in the absence of selective pressure to accommodate growth of antibiotic-sensitive target strains. The data underlying this Figure can be found in S1 Data.

environment. Our original goal was to test if gut colonization by *Pa* would exclude *Kp* from the gut, with a broader goal to evaluate the underpinning mechanisms as potential decolonization strategies. Yet, we found that *Pa* is not readily adaptable to our gut colonization model, which was confirmed in discussion with other groups that had attempted *Pa* gut colonization. Of note, we were unable to exactly replicate the experiments in S1A Fig with our wild *Pa* strains, as the source vendor of these mice no longer breeds C57Bl6 mice in the barrier (room of origin) from which the original mice were sourced. As the barrier of origin is a known confounder that impacts gut microbial community structure [66], it is challenging to specifically determine if our original observations are repeatable. However, our attempts to recapitulate these findings suggest that our original observation was happenstance, wherein *Pa* could have been a contaminant or transient gut colonizer, inhibiting *Kp* through a phenazine-independent modality, and/or another microbe or set of microbes was explanatory for the reduction in colonization that we observed in our original experiments. These accidental observations led to interesting findings, nonetheless.

Despite not identifying a microbial interaction that led to a reduction in *Kp* gut colonization, there are significant implications of these data for our understanding of how microbial interactions shape infection outcomes. In the lung, a growing number of studies recognize the importance of the lung microbial community in health outcomes, where it is associated with lung transplant failure, lung cancer, asthma, idiopathic pulmonary fibrosis, and infectious pneumonia, amongst others (reviewed in [2]). This concept is best studied in cystic fibrosis (CF). CF disease severity, as well as the immunological and treatment response, are all strongly influenced by microbial interactions between important pathogens, such as *Pa*, *Staphylococcus aureus*, and others (reviewed in [67,68]). During chronic *Pa* lung infection in CF patients, *Pa* adapts to a more cooperative state, resulting in reduction of phenazine expression [69–71]. In humans, lung infections appear to be exacerbated by *Pa-Kp* co-infection [11–13]. In a mouse model of lung co-infection, *Pa* exacerbated *Kp* infection, but the bacterial mechanisms underpinning that exacerbation are unknown [17]. In the urinary tract, it is well known that microbial interactions greatly shape infection outcomes, including disease severity and secondary effects of infection, such as formation of struvite crystals (reviewed in [72]). For example, *Providencia stuartii* enhances *Proteus mirabilis* dissemination during UTI, whereas *Morganella morganii* reduces *P. mirabilis* dissemination during UTI [73,74]. *Kp* and *Pa* urinary tract co-infections are known to occur [14], but little is known about what shapes their interactions and their infectious outcomes. Our data suggest there may be important impacts of phenazines on the local polymicrobial microenvironment during infection that have yet to be tested using these models; however, this requires further investigation. Interestingly, PYO is necessary for fitness during *Pa* lung infection [75]; thus, PYO (or other phenazines) may underpin the exacerbation of *Kp* infection by *Pa*. Conversely, RhlR (and therein, phenazines) is dispensable for fitness in a catheter-associate UTI mouse infection model using PA14 [76]. Thus, it may be that phenazines are important for shaping interference competition during UTI, rather than driving fitness during mono-microbial infection. Future studies should aim to disentangle the complex interactions between phenazines, the host, and other microbes during polymicrobial infection when *Pa* is present.

A mechanistic explanation for the site specificity we observe is that airway lining fluids are unusually antioxidant-rich, buffering phenazine-derived oxidants, whereas urinary/bladder matrices generally offer less antioxidant capacity. The epithelial lining fluid contains high glutathione and other low-molecular-weight antioxidants (ascorbate, urate, vitamin E), which can quench redox cycling [77,78]; this could dampen phenazine activity in BALF despite ample $O_2$. By contrast, measured human bladder urine $pO_2$ values are moderate (~23–45 mmHg) [79], which may be sufficient to sustain phenazine cycling and toxicity in bladder contexts. Our data suggest that *Pa*'s presence will not uniformly suppress *Kp* in the lung (where antioxidant buffering is high) but may restrain *Kp* in the urinary tract. This has at least three implications. First, knowing the electron-acceptor landscape could help predict when *Pa* will exclude *Kp* (or vice versa). Second, manipulating local redox (e.g., acceptor supply, iron availability) could potentiate or quell phenazine activity *in situ*, an approach complementary to antibiotic therapy. Third, because phenazines engage distributed physiological circuits, resistance may arise via network-level changes (efflux, redox enzymes) rather than single drug targets, arguing for rational combinations that close those escape valves in a site-specific manner. Together, these findings illustrate that environmental redox

chemistry and host-derived antioxidants shape whether bacterial competition manifests as antagonism or coexistence, an ecological principle likely generalizable across infection sites. These implications could refine risk prediction and empiric therapy choices in polymicrobial infection.

In other organ systems, microbial interactions have become key targets for health-oriented interventions. The gut is the most prominent focus of such efforts, including the successful use of fecal microbiota transplantation to treat recurrent *Clostridioides difficile* infection and ongoing clinical trials of live bacterial therapeutics (reviewed in [80]). Such interventions are at the nascent stage of consideration for both *Kp* and *Pa* infections, and microbial interactions are being considered as anti-infection therapeutic targets (reviewed in [81], for example). Our study shows that it may be necessary to consider the type of infection and strain genetic repertoire when evaluating the efficacy of curative or anti-infection therapies, as efficacy is likely site and strain specific. Additionally, our use of a heterologous overexpression system indicates that future studies should determine if engineered *Ec* or bacterial species with increased antimicrobial secondary metabolite production, such as PYO or PYR, could be used to eliminate *Kp* or other microbes in human-relevant systems. Moreover, by identifying phenazines as environment-conditioned interspecies effectors, our work reframes "one-microbe-one-toxin" narratives toward rules that consider chemistry, environment, and genotype.

There is an urgent need for new approaches to prevent deaths from antibiotic resistant bacterial infections as global antimicrobial resistance rises. Both *Kp* and *Pa* are important causes of antibiotic-resistant infection, both ranking in the top six most prevalent causes of deaths attributable to antibiotic resistance [24,25]. There are likely opportunities to shorten, prevent, or cure infections with these antibiotic-resistant pathogens through targeted manipulation of polymicrobial competitions that underpin and/or enhance their virulence. A major barrier in therapeutic leveraging of microbial competition is the lack of mechanistic studies of these interactions, which are necessary to understand and predict their efficacy. Additionally, a significant challenge for interventions that rely on microbial competition is the tremendous patient heterogeneity due to variations in genetics, immunological baseline, behavior, and microbial ecosystems (reviewed in [82]). As such, it is important to measure the effects of these variables when evaluating the efficacy of interventions that leverage microbial competition, when possible. Resulting principles may enable prediction of pathogen dynamics across host sites and inform development of ecological therapeutics that exploit interbacterial antagonism. If successful, such interventions have the potential to both lower the burden of infection complicated by antibiotic resistance and re-potentiate existing antibiotic therapies by reducing their use.

## Materials and methods

### Ethics statement

This study was performed in strict accordance with the recommendations in the *Guide for the Care and Use of Laboratory Animals* [83]. The Indiana University Institutional Animal Care and Use Committee approved this research (protocol #22114, PI: Jay Vornhagen). Wild *Pa* strains (145.1, 191.1, 193.1) were isolated from mouse experiments (S1A Fig) approved by The University of Michigan Institutional Animal Care and Use Committee (protocol #PRO00007474, PI: Michael A. Bachman). Clinical isolates were collected and de-identified by RFR in accordance with approval by the Indiana University Institutional Review Board (protocol #16139, PI: Ryan F. Relich).

### Materials, media, and bacterial strains

All chemicals were purchased from Sigma-Aldrich (St. Louis, MO) or Fisher Scientific (Fairlawn, NJ) unless otherwise indicated. Bacterial strains used in this study are described in Table 2. Bacteria were cultured in Luria-Bertani (LB) broth, or in M9 minimal medium (M9 salts, 0.2 M $MgSO_4$, 0.01 M $CaCl_2$, with 1% casamino acids, "M9-cas" or with 0.5% glucose, "M9-glu") at 37 °C with shaking at 220 rpm, or on LB agar at 27 °C (*Kp*) or 37 °C (*Pa* and *Ec*) supplemented with kanamycin (40 or 25 µg/mL) and/or rifampicin (30 µg/mL) as appropriate. pJL1-sfGFP was a gift from Michael Jewett (Addgene

**Table 2. Bacterial strains used in this study.**

| JV Number | Species | Strain | Source or reference |
|---|---|---|---|
| JV1 | Klebsiella pneumoniae | KPPR1 | [84] |
| JV3 | Pseudomonas aeruginosa | 145.1 | This study |
| JV4 | Pseudomonas aeruginosa | 191.1 | This study |
| JV5 | Pseudomonas aeruginosa | 193.1 | This study |
| JV10 | Klebsiella pneumoniae | 13F11 | [85] |
| JV11 | Escherichia coli | MG1655 | [86] |
| JV13 | Pseudomonas aeruginosa | PAO1 | [87] |
| JV14 | Pseudomonas aeruginosa | PA14 (used only for JV388 and JV389 comparisons) | Waters Lab, Michigan State University |
| JV25 | Enterococcus faecalis | NA | This study |
| JV282 | Klebsiella oxytoca | NA | This study |
| JV338 | Pseudomonas aeruginosa | PA14ΔrhlR | [48] |
| JV339 | Pseudomonas aeruginosa | PA14ΔrhlI | [48] |
| JV450 | Pseudomonas aeruginosa | PA14 (SMC232) | [88] |
| JV449 | Escherichia coli | DH5αpJL1-sfGFP | [89] |
| JV461 | Klebsiella pneumoniae | KPPR1pJL1-sfGFP | This study |
| JV462 | Pseudomonas aeruginosa | PA14ΔphzA1-G1ΔphzA2-G2 ("PA14ΔphzA-G," SMC5020) | [90] |
| JV463 | Pseudomonas aeruginosa | PA14 SXO phzS ("PA14ΔphzS," SMC5123) | [91] |
| JV466 | Pseudomonas aeruginosa | PA14ΔphzHΔphzM SXO phzS (SMC5126) | [91] |
| JV467 | Pseudomonas aeruginosa | PA14ΔphzH (SMC5127) | [91] |
| JV468 | Pseudomonas aeruginosa | PA14ΔphzM (SMC5128) | [91] |
| JV418 | Escherichia coli | UTI89 | [92] |
| JV531 | Escherichia coli | SJ102 (MG1655 intC::λpR-YFP-Cmr) + pSB3K3_EVC ("pEmpty") | This study |
| JV534 | Escherichia coli | SJ102 + pSB3K3_phzA-GSM ("pPhzA-G") | [93] |
| JV537 | Escherichia coli | SJ102 + pSB3K3_phzA-GS*M ("pPhzA-GS*M") | This study |
| JV540 | Escherichia coli | SJ102 + pSB3K3_phzA-GSM ("pPhzA-GSM") | [93] |
| JV543 | Klebsiella oxytoca | NA | This study |
| JV659 | Escherichia coli | CTF073-GFP | [94] |

plasmid 102634). The PYR expression plasmid (pSB3K3_*phzA-GS*M*) was obtained spontaneously by serially plating on LB-agar, leading to the identification of a colony which displayed a red corona. The plasmid was subsequently purified and re-transformed into a fresh *E. coli* SJ102 background strain to avoid any genomic mutations. pSJ102 containing *Ec* was grown at 30 °C for optimal phenazine production. For experiments that required species-specific selection, KPPR1 was selected on LB agar with 30 μg/mL rifampicin grown at 27 °C and *Pa* was selected on Pseudomonas Isolation Agar (Becton, Dickinson and Company, Franklin Lakes, NJ). Anaerobic growth was performed in a Coy Lab's Vinyl Anaerobic Chamber.

### Phenazine synthesis plasmids

Plasmids for PCA (pSB3K3_*phzA-G*) and PYO (pSB3K3_*phzA-GSM*) expression were obtained from a previous study [93]. The empty vector control (pSB3K3_EVC) was made by performing a NotI digestion of pSB3K3_*phzA-GSM*, followed by ligation with a neutral oligonucleotide linker made by annealing the following primers: 5′-GGCCGCGTACGACT TACGC-3′ and 5′-GGCCGCGTAAGTCGTACGC-3′. All plasmids were transformed into *E. coli* SJ102 (MG1655 intC::λpR-YFP-Cmr [95]) by electroporation. The PYR expression plasmid (pSB3K3_*phzA-GS*M*) was obtained spontaneously by

selecting a colony of pSB3K3_*phzA-GSM* cells which displayed a red corona on LB-agar plates. Plasmid sequences were confirmed by full plasmid sequencing.

## Whole genome sequencing

*Pa* and *Ec* strains were cultured overnight in LB broth with appropriate antibiotics for 24 hours prior to DNA extraction. DNA was extracted using Monarch Genomic DNA Purification Kit (New England Biolabs, Ipswich, MA) according to the manufacturer's instructions with the only deviation being the substitution of 60 °C elution buffer for 34 °C DNase/RNase-Free Water (Zymo Research, Irvine, CA). The extracted DNA quantity was analyzed using a Qubit 4 Fluorometer 1× dsDNA HS analysis. DNA extractions above a concentration of 8 ng/uL were prepared for sequencing with a Rapid Barcoding Kit 24 V14 (Oxford Nanopore Technologies, Oxford, England) utilizing their Rapid sequencing DNA V14 – barcoding protocol (RBK_9176_v114_revQ_27Dec2024). Bacterial genomes were sequenced with a Flo-Min114 flow cell on a MinION Mk1B device (Oxford Nanopore Technologies, Oxford, England) and base-called using the ONT MinKNOW software v24.06.5 (Oxford Nanopore Technologies, Oxford, England). The reads were assembled into scaffolds using Flye Assembler v2.9.5 [96].

## Phylogenetic analysis

We determined that the in-house assembled genomes were most similar to ST175 by multi-locus sequence typing [97]. These assembled genomes were combined with genomic sequences of five ST175 *Pa* strains downloaded from the bacterial and viral bioinformatics resource center on December 9, 2023 [98]. The genomic sequences of an additional 65 *Pa* strains were downloaded from the National Center for Biotechnology Information on December 6, 2023 [99]. These 73 genomes, in addition to the three in-house assembled genomes, were annotated using Prokka v1.14.5 [100] and the core genome determined using Roary v3.13.0 [101]. The core genome in this study was defined as loci with 90% consistency between isolates. A maximum likelihood phylogenetic tree was assembled using FastTree v2.1.11 and visualized using iTOL v7.0 [102,103].

## Interspecies competition assays

For co-culture competition assays, strains were cultured overnight LB or M9-cas. Strains were then co-inoculated 1:1,000 in 1 mL of the medium of interest and mixed. This input was serially diluted and selectively spot-plated for CFU quantification. After 24 hours of shaking incubation, the output was again serially diluted and plated. For spent-media competition assays, strains were cultured overnight in the medium of interest. Spent media was created by filtering the supernatant of centrifuged culture with a 0.22 μm syringe filter. The strain of interest was inoculated 1:1,000 into spent media and was then plated and incubated as described above. For MG1655pPhzA-GS*M spent media assays, except those in Fig 3F, dithiothreitol was added to final concentration of 5 mM.

## Growth curves

Strains were cultured overnight in culture medium of interest (LB or M9-cas) then diluted to an $OD_{600}$ of 0.01 in the medium of interest the following day. Cultures were incubated at 37 °C with aeration and $OD_{600}$ readings were taken every 15 min using a BioTek microplate reader with Gen5 software (Version 3.12.08, BioTek, Winooski, VT) for 24 hours. To simultaneously assess doubling time, growth rate, lag time, non-sigmoidal growth due to stress and bacterial density, area under the curve analysis was used to quantify differences in growth, as in [38].

## BioLog Phenotype MicroArray analysis

BioLog Phenotype MicroArrays (Biolog, Hayward, CA) were performed according to manufacturer's instructions with some modifications, as in [38]. KPPR1, PAO1, PA14, and the mouse-derived *Pa* strains (*Pa* 145.1, *Pa* 191.1, and *Pa* 193.1),

were cultured overnight in LB, then bacteria were pelleted, washed once in sterile PBS, and re-suspended in sterile PBS to avoid aberrant transfer of xenonutrients. Each strain was diluted in IF-0 medium to a final $OD_{600}$ of 0.035 and 100 µL was plated onto plates PM1 and PM2 with gentle mixing. After inoculation, plates were statically incubated overnight at 37 °C. Following 24 hours of incubation, growth was measured at $OD_{595}$.

## PA14NR library screen

Construction of the PA14NR transposon library used in this study has been extensively described elsewhere [104]. Library plates were pin-replicated onto black 96-well microplates containing 100 µL of LB agar. After 24 hours of static growth at 37 °C, liquid culture of KPPR1pJL1-sfGFP was diluted 1:1,000 in LB broth and 100 µL was added on top of the agar. After 24 hours of static 37 °C incubation, a fluorescence read was performed (Excitation: 479, Emission: 520, Optics: Bottom, Gain: 50).

## High-resolution mass spectrometry

For high-resolution mass spectrometry (HR-MS) detection of phenazines in cell-free supernatant, bacteria were grown overnight in LB broth at 30 °C or 37 °C with cognate antibiotics. Cultures were centrifuged at 7,000$g$ for 5 min and supernatant was filtered through a 0.22 µm syringe filter. This filtrate was then centrifuged at 16,000$g$ for 10 min to remove debris, and 500 µL was diluted 1:1 with LC-grade methanol. Pure phenazines were also included. Sample (1 µL) was injected onto 1290 LC - 6545 QTOF. The ESI-HRMS analyses were performed in positive ion mode, utilizing nitrogen as the nebulizing and drying gas. The instrumental conditions were as follows: nebulizer pressure, 25 psi; drying temperature, 325 °C; drying gas, 8 L/min; sheath gas temperature, 400 °C; sheath gas flow, 12 L/min; fragmentor, 220 V; skimmer, 65 V; Oct 1 RF Vpp, 750 V. For the first HPLC condition, an Agilent ZORBAX Eclipse Plus C18 column (HD 2.1*5 0mm 1.8-Micron) was used for separation. 0.1% formic acid in water (v:v) as mobile phase A and 0.1% formic acid acetonitrile as mobile phase B (v:v) were used. The flow rate is 0.6 mL/min. The gradient starts as 5% B and holds for 0.5 min, increased from 5%B to 95%B in 4.5 min and holds for 0.5-min, 95%B to 5%B in 0.5 min and holds for 0.5 min. For the second HPLC condition and ACQUITY UPLC BEH Amide column (150 mm × 2.1, 1.7 µm) was used for separation. The mobile phases were heated to 45 degrees. 0.1% formic acid in water (v:v) as mobile phase A and 0.1% formic acid acetonitrile as mobile phase B (v:v) were used. The flow rate is 0.6 mL/min. The gradient starts as 5% A and holds for 1 min, increased from 5% A to 95% A in 9 min and holds for 1 min, 95% A to 5% A in 1 min and holds for 1 min. Data were analyzed using mzmine [105]. Extracted ion chromatographs were generated for PCA ([M-OH]+ = 207.0372 m/z; [M-COOH]+ = 179.0456 m/z), 5MPCA ([M+] = 239.0815 $m/z$), and PYO ([M+] = 211.0866 $m/z$) with 10 ppm error.

## MIC and MBC assays

MIC assays were performed as previously described [106]. For PCA, 1-HP, and PYO (1-phenazinecarboxylic acid, 1-phenazinol, 5-methyl-1(5H)-phenazinone, respectively, Cayman Chemical, Ann Arbor, Michigan) was resuspended to a concentration of 2–16 mg/mL in dimethylsulfoxide and then diluted into LB, M9-glu +/− 10 mM $NaNO_3$, or PA14Δ*phzA-G* spent media to 1 mg/mL. 100 µL of this solution was plated into U bottom 96-well plate in triplicate and then 2-fold serially diluted into LB or PA14Δ*phzA-G* spent media 10 times. Then, 50 µL of KPPR1 culture inoculated in LB or PA14Δ*phzA-G* spent media to final concentration of $OD_{600}$ = 0.02 was plated across each serial dilution, yielding a final $OD_{600}$ = 0.01, and incubated at 37 °C shaking for 24 hours. After 24 hours, MIC was determined to be the lowest concentration where growth was not visually observed.

For PYO MBC assays, the entire MIC assay described above was spot-plated onto LB agar at 24 hours and grown overnight at 37 °C. After overnight growth, MBC was determined to be the lowest concentration at which no *Kp* colonies were recovered.

For PYR, 100 μL of 100% MG1655pPhzA-GS*M spent media was plated into U bottom 96-well plate and serially diluted in MG1655pEmpty spent media with 10 mM DTT 7 times. Then, 50 μL of KPPR1 culture inoculated in MG1655pEmpty spent media to final concentration of $OD_{600} = 0.02$ was plated across each serial dilution, yielding a final DTT concentration of 5 mM and $OD_{600} = 0.01$. Additionally, KPPR1 culture was inoculated in 100 μL of 100% MG1655pPhzA-GS*M spent media with 5 mM DTT to final $OD_{600} = 0.01$ to ensure a 100% MG1655pPhzA-GS*M spent media condition. All media were incubated at 37 °C shaking for 24 hours, and MIC was determined as above.

To assess additive and/or synergistic effects of PYO and PYR, a checkerboard assay was used. To this end, a 2-fold serial dilution of PYO, starting at 62.5 μg/mL was combined with 2-fold serial of MG1655pPhzA-GS*M spent media, starting at 100%. DTT was added to a final concentration of 5 mM and KPPR1 culture was inoculated to final $OD_{600} = 0.01$. This array was incubated at 37 °C shaking for 24 hours. After 24 hours, MIC was determined to be lowest concentration where growth was not visually observed.

### Phenazine quantification

The absorbance spectra of pure PYO and MG1655pPhzA-GS*M spent media was measured in both reducing (5 mM DTT) and oxidizing conditions (0.3% peroxide). We determined that oxidized MG1655pPhzA-GS*M spent media and untreated PYO yielded the greatest differences between these spectra at $Abs_{500}$ and $Abs_{695}$, respectively. For PYO quantification, *Pa* cells were removed from spent *Pa* media generated under conditions identical to spent media assays. The raw $Abs_{695}$ of spent media was measured or compared to a standard curve of pure PYO to estimate PYO concentrations. For PYR quantification, *Pa* spent media was oxidized, then the raw $Abs_{500}$ of spent media was measured or compared to a standard curve of oxidized MG1655pPhzA-GS*M spent media to interpolate estimate PYR concentrations.

### Collection and screening of clinical isolates

The IU Health Division of Clinical Microbiology collected clinical isolates between May and July 2023 from IU Health facilities across the state of Indiana. Of these, 198 isolates were identified by MALDI-TOF (Bruker) as *Kp,* and 194 as *Pa.* These isolates were derived from a variety of infection sites and patient information was de-identified. Isolates were passaged once from primary plating media (sheep blood agar or MacConkey agar) prior to storage on brain-heart infusion agar slants (ThermoFisher) at room temperature.

To screen *Kp* growth in PA14 spent media, clinical *Kp* isolates and KPPR1 were cultured overnight in LB and inoculated 1:100 into sterile spent PA14 media or fresh LB. After 24 hours of shaking incubation at 37 °C, *Kp* growth was measured at $OD_{600}$.

To screen *Kp* growth against specific phenazines, a subset of 49 clinical *Kp* isolates and KPPR1 were cultured in sterile spent MG1655pEmpty media. Eight isolates were determined to be unable to grow in this spent media and excluded from future assays. The remaining 41 clinical *Kp* isolates were cultured in sterile spent MG1655pPhzA-G, MG1655pPhzA-GS*M, or MG1655pPhzA-GSM media generated in the absence of antibiotic selection for their respective expression plasmids to permit clinical isolate growth. After 24 hours of shaking incubation at 37 °C, *Kp* growth was measured at $OD_{600}$.

To screen clinical *Pa* isolate restriction of KPPR1 growth, clinical *Pa* isolates and KPPR1pJL1-sfGFP were cultured overnight in LB and inoculated 1:1,000 in LB for a 1:1 competition of KPPR1pJL1-sfGFP in co-culture with one clinical isolate. After 24 hours of shaking incubation at 37 °C, an $OD_{600}$ reading was taken, as was a fluorescence read (Excitation: 479, Emission: 520, Optics: Top, Gain: 50).

### *Ex vivo* interspecies competition assays

Strains for *ex vivo* interspecies competition assays were cultured overnight in LB, washed in sterile PBS and re-suspended in sterile PBS before inoculation to avoid aberrant transfer of xenonutrients.

*Large intestinal contents:* 6-to-8-week-old C57Bl6/J male mice were sourced from the Jackson Labs from barrier RB07. Following acclimation, mice were euthanized, the large intestine (cecum and colon) was removed, and its contents were gently homogenized in 2 mL sterile, pre-reduced PBS and divided into 400 μL aliquots. Immediately following collection, samples were transferred to an anaerobic chamber and inoculated to a final concentration of ~$5 \times 10^7$ CFU KPPR1 alone or an equal ratio of KPPR1 with each wild *Pa* strain. Samples were incubated at 37 °C for 48 hours, then dilution plated on species-selective media to quantify final bacterial densities. Only male mice were used to account for sex-derived microbiome differences.

*BALF:* 6-to-8-week-old C57Bl6/J male mice from the Jackson Labs' barrier RB07 and C57Bl6 male mice from Charles River from barrier K61 were sourced for BALF collection. Two vendor-barrier sources were used to determine if there were vendor-specific effects of the lung microbiome, as has been previously reported [107]. Following acclimation, mice were euthanized, and BALF was collected as previously described [38]. If BALF recovery was >1 mL, sterile PBS was added to 1 mL. BALF was divided into 400 μL aliquots and inoculated 1:1,000 with a 1:1 mix of KPPR1 and WT PA14 or PA14Δ*phzA-G*. Input bacterial densities were determined by dilution plating on species-selective media. Samples were incubated aerobically with agitation (220 rpm) at 37 °C for 24 hours, then dilution plated on species-selective media to quantify final bacterial densities. Only male mice were used to account for sex-derived microbiome differences, and no vendor-based differences were detected.

*Bladder homogenate:* Bladders were collected from the mice used for BALF collection. Bladders were collected into 1 mL sterile PBS, then thoroughly homogenized. Homogenized tissue was centrifuged at 21,300*g* and resulting supernatant was collected. If homogenate recovery was 1 mL, sterile PBS was added to 1 mL. Bladder homogenate was divided into 400 μL aliquots and inoculated 1:1,000 with a 1:1 mix of KPPR1 and WT PA14 or PA14Δ*phzA-G*. Input bacterial densities were determined by dilution plating on species-selective media. Samples were incubated aerobically with agitation (220 rpm) at 37 °C for 24 hours, then dilution plated on species-selective media to quantify final bacterial densities. As above, no vendor-based differences were detected.

## In vivo models

*Gut colonization model:* Gut colonization was performed as previously described [38,66], with some minor modifications. Briefly, 6-to-8-week-old C57Bl6/J (equal numbers of male and female) mice were sourced from the Jackson Labs from barrier RB07. Following acclimation, prior to colonization, mice were administered 0.5 g/L ampicillin via drinking water. Four days following antibiotic administration, mice were orally gavaged with ~$10^7$ CFU stationary-phase *Pa* 191.1 suspended in 250 μL sterile PBS. Fecal pellets were collected at predetermined timepoints, homogenized in 300 μL sterile PBS, and *Pa* gut density was measured via dilution plating on selective medium. Seven days after 191.1 administration, mice were orally gavaged with ~$10^8$ CFU stationary-phase KPPR1 suspended in 250 μL sterile PBS. Fecal pellets were collected at predetermined timepoints, homogenized in 300 μL sterile PBS, and *Kp* gut density was measured via dilution plating on selective medium. Mice were euthanized seven days after KPPR1 administration, ceca were collected and homogenized in 1 mL sterile PBS, and *Kp* density was measured via dilution plating on selective medium.

## Statistical analysis

All *in vitro* experimental replicates represent biological replicates. For in vitro studies two-tailed Student *t*-test or ANOVA followed by Tukey's multiple comparisons post-hoc test on $\log_{10}$ transformed data (transformation performed to fit data to normal distribution) was used to determine significant differences between groups. For *ex vivo* and in vivo studies, all experiments were replicated at least twice, accounting for sex as a biological variable when appropriate. A *p*-value of less than 0.05 was considered statistically significant for the above experiments, and analysis was performed using base R version 4.5.0. Data were processed, analyzed, and visualized using R packages "readxl," "tidyverse," "dplyr," "vegan," "outliers," "EnvStats," "ggplot2," "ggrepel," and "ggtext."

## Use of AI

ChatGPT-4o was used to edit the R scripts associated with this manuscript to enhance their accessibility following human-driven drafting and analysis. Functionality of all edited scripts was confirmed prior to publication.

## Supporting information

**S1 Fig. Detection and phylogenetic characterization of wild *Pa* strains in *Kp*-colonized mice.** 6- to 8-week-old C57 mice from Taconic Farms were orally inoculated with $10^8$ CFU WT KPPR1 or 13F11. *Kp* was enumerated from feces at 24 hours and large intestinal contents (LI) 48 hours post-inoculation (**A**). *Pa* was isolated from mouse 145, 191, and 193 and subjected to whole genome sequencing. These strains and subset of *Pa* isolates from a previous study [43] were used to build an approximately-maximum-likelihood phylogenetic tree based on a core genome alignment of these strains to determine if strains 145.1, 191.1, and 193.1 group with Clade A (PAO1 clade), Clade B (PA14 clade), or Clade C (split Pa7 clade). Select *Pa* strains absent in the previous study that represent ST175 were also included, as strains 145.1, 191.1, and 193.1 were predicted to be most like ST175 using multi-locus sequence typing. The ST175 representative and mouse strains grouped with Clade A, thus they are labeled "Proposed Clade A" (**B**). The data underlying this Figure can be found in S2 Data and the raw phylogenetic tree data can be found in S4 Data.
(TIF)

**S2 Fig. *Ec* MG1655 and an isogenic neutral Tn mutant do not restrict KPPR1 growth in LB.** *Kp* KPPR1 was grown alone or in co-culture in LB with *Ec* MG1655 (**A**), *Kp* 13F11 (**B**), or in filter sterilized spent media of KPPR1, *Ec* MG1655 (**C**), or *Kp* 13F11 strain (**D**). For **A**–**D**, "Log fold change$_{24\ hours}$" = $\log_{10}$(output KPPR1 CFU at 24 hours/input KPPR1 CFU). *p*-values represent Tukey multiple comparison correction following one-way ANOVA. Each data point is a biological replicate, and horizontal lines indicate the mean of each dataset. The data underlying this Figure can be found in S2 Data.
(TIF)

**S3 Fig. *Pa* restricts *Kp* growth in M9 medium supplemented with 1.0% casamino acids in a strain-dependent, contact-independent manner.** *Kp* KPPR1 was grown alone or in co-culture in M9 medium supplemented with 1.0% casamino acids with mouse-derived wild *Pa* (**A**), PAO1, PA14 (**B**) or in filter-sterilized spent media of KPPR1 or each *Pa* strain (**C**–**D**). For **A**–**D**, "Log fold change$_{24\ hours}$" = $\log_{10}$(output KPPR1 CFU at 24 hours/input KPPR1 CFU). *p*-values represent Tukey multiple comparison correction following one-way ANOVA. Each data point is a biological replicate, and horizontal lines indicate the mean of each dataset. The data underlying this Figure can be found in S2 Data.
(TIF)

**S4 Fig. *Ec* MG1655 and an isogenic neutral Tn mutant do not restrict KPPR1 growth in M9 medium supplemented with 1.0% casamino acids.** *Kp* KPPR1 was grown alone or in co-culture in M9 medium supplemented with 1.0% casamino acids with *Ec* MG1655 (**A**), *Kp* 13F11 (**B**), or in filter-sterilized spent media of KPPR1, *Ec* MG1655 (**C**), or *Kp* 13F11 strain (**D**). For **A-D**, "Log fold change$_{24\ hours}$" = $\log_{10}$(output KPPR1 CFU at 24 hours/input KPPR1 CFU). *p*-values represent Tukey multiple comparison correction following one-way ANOVA. Each data point is a biological replicate, and horizontal lines indicate the mean of each dataset. The data underlying this Figure can be found in S2 Data.
(TIF)

**S5 Fig. Growth dynamics of select *Kp, Pa,* and *Ec* strains in LB and M9 medium supplemented with 1.0% casamino acids.** KPPR1, PAO1, PA14, *Pa* 145.1, *Pa* 191.1, *Pa* 193.1, MG1655, 13F11, PA14Δ*rhlR*, PA14Δ*rhlI* were grown in LB (**A-E**) or in M9 medium supplemented with 1.0% casamino acids (**F-J**). Area under the curve (AUC) analysis was used to quantify growth at early (0–6 hours, **A-B, F-G**) and late (0–24 hours, **A-J**) stages of growth. *p*-values represent Tukey multiple comparison correction following one-way ANOVA. For growth curves, each data point represents the mean, and

vertical bars represent the standard error of the mean. For AUC analysis, each data point is a biological replicate, and horizontal lines indicate the mean of each dataset. The data underlying this Figure can be found in S2 Data.
(TIF)

**S6 Fig. WT *Pa* and *Ec* strains are less metabolically flexible than *Kp.*** KPPR1, PAO1, PA14, *Pa* 145.1, *Pa* 191.1, *Pa* 193.1, and *Ec* MG1655pEmpty were grown in BioLog Phenotype Microarray plates PM1 and PM2 (mean of three biological replicates displayed, each row is an individual carbon source, **A**). Euclidean distance was used to measure the dissimilarity between the growth phenotypes of each strain (**B**). The data underlying this Figure can be found in S2 Data.
(TIF)

**S7 Fig. PA14NR library screen and validation.** To identify candidate factors involved in *Kp* growth restriction, the PA14NR library was replicate plated onto LB-agar, grown overnight at 37 °C, and inoculated with GFP-expressing KPPR1 (**A**). After 24 hours of co-culture, the fluorescence of each co-culture was measured, yielding 18 candidate transposon mutants, 16 of which exhibited reduced restriction, and 2 that exhibited enhanced restriction. PA14 and the 18 candidate transposon mutants identified to have a role in *Kp* growth restriction were grown in LB (**B**) and area under the curve (AUC) analysis was used to quantify growth (**C**). In panel **B**, each data point represents the mean, and vertical bars represent the standard error of the mean. *Kp* KPPR1 was grown alone or in co-culture in LB with PA14 or the 18 transposon mutants (**D**) or in filter-sterilized spent media of KPPR1, PA14, or the 18 transposon mutants (**E**). PYO (**F**) and PYR (**G**) were measured at 695 and 500 nm, respectively, from select transposon mutants. For **D**–**E**, "Log fold change from WT PA14" = $\log_{10}$(output KPPR1 CFU at 24 hours/input KPPR1 CFU) in transposon mutant co-culture or spent media culture/ $\log_{10}$(output KPPR1 CFU at 24 hours/input KPPR1 CFU) in WT PA14 co-culture or spent media culture. For **C**, *p*-values represent Tukey multiple comparison correction following one-way ANOVA, and for **D**–**E**, *p*-values represent one-sample *t* test from a hypothetical mean of 0. For **C**–**G**, each data point is a biological replicate, horizontal lines indicate the mean of each dataset, and in **C**–**E**, red datasets are statistically significant from their relative comparisons. Panel **A** was Created in BioRender. Tilston-lunel, N. (2026) https://BioRender.com/6b5vwn8. The data underlying this Figure can be found in S2 Data.
(TIF)

**S8 Fig. *rhlRI* is required for *Kp* growth restriction by *Pa* in M9 medium supplemented with 1.0% casamino acids.** KPPR1 was grown alone or in co-culture in M9 medium supplemented with 1.0% casamino acids with WT PA14, PA14Δ*rhlR*, or PA14Δ*rhlI* (**A**) or in filter-sterilized spent media of KPPR1 or each *Pa* strain (**B**), supplemented with water, 1% casamino acids ("Cas," **C**) or 0.4% glucose ("Glu," **D**). For **A-D**, "Log fold change$_{24\text{ hours}}$" = $\log_{10}$(output KPPR1 CFU at 24 hours/input KPPR1 CFU). *p*-values represent Tukey multiple comparison correction following one-way ANOVA. *p*-values over columns indicate comparison to "Fresh media" condition. Each data point is a biological replicate, and horizontal lines indicate the mean of each dataset. The data underlying this Figure can be found in S2 Data.
(TIF)

**S9 Fig. PYR activity is redox-dependent.** 13F11 (Kan$^R$ KPPR1 variant) was grown in fresh LB broth or spent media from MG1655 constitutively expressing PYR (pPhzA-GS*M). Dithiothreitol (DTT) was titrated into both media. "$\log_{10}$(Fold change from no DTT)" = $\log_{10}$(output 13F11 CFU at 24 hours/input 13F11 CFU) in fresh or spent media + DTT/ $\log_{10}$(output 13F11 CFU at 24 hours/input 13F11 CFU) in fresh or spent media without DTT. Each data point is a biological replicate, and horizontal lines indicate the mean of each dataset. The data underlying this Figure can be found in S2 Data.
(TIF)

**S10 Fig. *Pa phzS* mutants do not produce PYR and PA14, and heterologous phenazine-expressing *Ec* do not produce 1-HP.** LC-MS was used to quantify PYR secretion from WT PA14, MG1655pPhzA-GS*M, PA14Δ*phzS*, and PA14_09400, which is a *phzS* transposon mutant from PA14NR library (**A**) or 1-HP secretion from WT PA14,

MG1655pEmpty, MG1655pPhzA-G, MG1655pPhzA-GS*M, or MG1655pPhzA-GSM in comparison to pure 1-HP (**B**). The data underlying this Figure can be found in S2 Data.
(TIF)

**S11 Fig. Clinical *Pa* strain phenazine production.** The spectra of pure PYO and the spent media MG1655 constitutively expressing PYR (pPhzA-GS*M) was measured under native, oxidizing (0.3% hydrogen peroxide) and reducing (5 mM dithiothreitol [DTT]) conditions (**A**) to identify wavelengths at which those phenazines can be differentiated. We determined that PYO and PYR in oxidizing conditions can be differentiated at 695 and 500 nm, respectively (red vertical bars). Clinical *Pa* strains (N = 194), WT PA14, and PA14Δ*phzA-G* were grown in LB broth and the $Abs_{695}$ of spent media was measured after 24 hours (**B**). $Abs_{695}$ results were correlated to growth restriction results from **Fig 5C** (Spearman correlation test, **C**). Clinical *Pa* strains (N = 192), WT PA14, PA14Δ*phzA-G*, and MG1655 constitutively expressing PYR (pPhzA-GS*M) and PYO (pPhzA-GSM) were grown in LB broth and the $Abs_{500}$ of spent media was measured after 24 hours following oxidation by 0.3% hydrogen peroxide (**D**). $Abs_{500}$ results were correlated to growth restriction results from **Fig 5C** (Spearman correlation test, **E**). $Abs_{500}$ and $Abs_{695}$ results were correlated with one another (Spearman correlation test, **F**). Each data point represents the mean read of each culture. The data underlying this Figure can be found in S2 Data.
(TIF)

**S12 Fig. Phenazines do not restrict *Kp* growth under anaerobic conditions.** KPPR1 was grown anaerobically alone or in co-culture in LB with mouse-derived wild *Pa*, PAO1, and PA14 (**A**). 13F11 (Kan$^R$ KPPR1 variant) was grown anaerobically in filter-sterilized spent media of MG1655 containing an empty vector (pEmpty) or constitutively expressing PCA (pPhzA-G), PYR (pPhzA-GS*M), and PYO (pPhzA-GSM, **B**). The minimum inhibitory concentration (MIC) and minimum bactericidal concentrations (MBC) were determined for KPPR1 for pure PYO in M9 minimal media with 0.5% glucose and 10 mM NaNO$_3$ (**C**). For **A-B**, "Log fold change$_{24 hours}$" = $\log_{10}$(output *Kp* CFU at 24 hours/input *Kp* CFU). Each data point is a biological replicate; horizontal lines indicate the mean of each dataset. For **C**, *p*-values represent Tukey multiple comparison correction following one-way ANOVA, comparing aerobic versus anaerobic MICs within each phenazine. Note that 13F11 growth in MG1655 spent media is no different than self-spent media in anaerobic conditions, whereas growth is restricted in the presence of PYR and PYO in aerobic conditions (see **Fig 3F**). The data underlying this Figure can be found in S2 Data.
(TIF)

**S13 Fig. Validation of clinical *Kp* screen results.** Clinical *Kp* strains (*N* = 6) were selected for validation (**A**). KPPR1 or select *Kp* strains were grown alone or co-cultured with select clinical *Pa* strains (**B**). For **B**, "Log fold change$_{24 hours}$" = $\log_{10}$(output KPPR1 CFU at 24 hours/input KPPR1 CFU). *p*-values represent Tukey multiple comparison correction following one-way ANOVA compared to the "PA14 + KPPR1" condition. Each data point is a biological replicate, and horizontal lines indicate the mean of each dataset. The data underlying this Figure can be found in S2 Data.
(TIF)

**S14 Fig. Validation of clinical *Pa* screen results.** Clinical *Pa* strains (*N* = 10) were selected for validation (**A**). KPPR1 was grown alone or in co-culture with select clinical *Pa* strains (**B**). For **B**, "Log fold change$_{24 hours}$" = $\log_{10}$(output KPPR1 CFU at 24 hours/input KPPR1 CFU). *p*-values represent Tukey multiple comparison correction following one-way ANOVA compared to the "KPPR1 alone" condition. Each data point is a biological replicate, and horizontal lines indicate the mean of each dataset. The red line represents the mean normalized fluorescence for KPPR1, and the orange line represents the mean normalized fluorescence for KPPR1 + PA14. The data underlying this Figure can be found in S2 Data.
(TIF)

**S15 Fig. *Kp* growth restriction in clinical *Pa* spent media is variable.** KPPR1 was grown in fresh LB medium or filter-sterilized spent media of KPPR1, PA14, JV46, or JV69 (**A**). "Log fold change$_{24 hours}$" = $\log_{10}$(output KPPR1 CFU at 24

hours/input KPPR1 CFU). Datapoints outlined in red are below the limit of detection (200 CFU/mL). $p$-values represent Tukey multiple comparison correction following one-way ANOVA. Each data point is a biological replicate, and horizontal lines indicate the mean of each dataset. LC-MS was used to quantify phenazine secretion from JV46 (**B**). The data underlying this Figure can be found in S2 Data.
(TIF)

**S16 Fig.** *Pa* **does not exclude** *Kp* **from the gut.** C57Bl6/J mice were treated for 4 days with 0.5 g/L ampicillin, then orally gavaged with ~$10^7$ CFU 191.1. Seven days post-191.1 colonization, mice were orally gavaged a second time with ~$10^8$ CFU KPPR1 ($N = 10$). Antibiotic-treated KPPR1 mono-colonized mice served as a control ($N = 10$). 191.1 fecal loads were monitored post-colonization (**A**), and KPPR1 fecal (**B**) and cecal loads (**C**) were measured were monitored post-colonization (after 7 days 191.1 colonization) or at the end of the experiment, respectively. Large intestinal contents from C57Bl6/J (no antibiotic treatment, $N = 5$) mice were collected and resuspended in sterile PBS. ~$5 \times 10^7$ CFU KPPR1 alone or an equal ratio of KPPR1 with each wild *Pa*, was inoculated into large intestinal contents and grown anaerobically. KPPR1 density was measured at 48 hours (**D**). The data underlying this Figure can be found in S2 Data.
(TIF)

**S17 Fig.** *Pa* **is viable in** *ex vivo* **tissues.** *Pa* density was measured in large intestinal contents (**A**), BALF (**B**), and bladder homogenate (**C**) from experiments presented in **Figs S15A**, **6B**, and **6C**, respectively. The data underlying this Figure can be found in S2 Data.
(TIF)

**S1 Table. Clinical** *Pa* **isolation sources.**
(XLSX)

**S1 Data. Raw data for main body figures.**
(XLSX)

**S2 Data. Raw data for supplemental figures.**
(XLSX)

**S3 Data. Complete PA14NR screen data.**
(XLSX)

**S4 Data. Tree file for S1B Fig.**
(TRE)

## Acknowledgments

The authors would like to acknowledge Dr. Natasha Tilston and the members of her lab for their thoughtful insight into this project even though they are virologists. The authors also acknowledge Dr. Kelly Bachta for sharing their protocols and insight into in vivo *P. aeruginosa* experiments. The authors thank Professor Christopher J. Howe and Dr. Robert W Bradley for providing phenazine production plasmids. Finally, the authors acknowledge the Indiana University Pervasive Technology Institute for providing supercomputing resources that have contributed to the research results reported within this paper.

## Author contributions

**Conceptualization:** Dominique H. Limoli, Christopher Whidbey, Jay Vornhagen.

**Data curation:** Jay Vornhagen.

**Formal analysis:** Katlyn Todd, Olivia Schneider, Christopher Whidbey, Jay Vornhagen.

**Funding acquisition:** Jay Vornhagen.

**Investigation:** Katlyn Todd, Olivia Schneider, Josefina L. Aronoff, Valerie Velázquez-Colón, Verónica Santana-Ufret, Nicole L. Anderson, Krista Gunter, Moraima Noda, Christopher Whidbey, Jay Vornhagen.

**Methodology:** Bartosz Witek, Lifan Zeng, Christopher Whidbey, Jay Vornhagen.

**Project administration:** Jay Vornhagen.

**Resources:** Joshua M. Lawrence, Ryan F. Relich, Lifan Zeng, Dominique H. Limoli, Jay Vornhagen.

**Supervision:** Jay Vornhagen.

**Validation:** Jay Vornhagen.

**Visualization:** Jay Vornhagen.

**Writing – original draft:** Katlyn Todd, Olivia Schneider, Joshua M. Lawrence, Christopher Whidbey, Jay Vornhagen.

**Writing – review & editing:** Katlyn Todd, Olivia Schneider, Joshua M. Lawrence, Josefina L. Aronoff, Bartosz Witek, Valerie Velázquez-Colón, Verónica Santana-Ufret, Moraima Noda, Ryan F. Relich, Dominique H. Limoli, Christopher Whidbey, Jay Vornhagen.

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
