## [Editor Report · Decision Letter 0]

29 Oct 2025

Dear Dr Vornhagen,

Thank you for submitting your manuscript entitled "Environmental redox conditions tune phenazine-mediated bacterial antagonism" for consideration as a Research Article by PLOS Biology.

Your manuscript has now been evaluated by the PLOS Biology editorial staff as well as by an academic editor with relevant expertise and I am writing to let you know that we would like to send your submission out for external peer review.

Once your full submission is complete, your paper will undergo a series of checks in preparation for peer review. After your manuscript has passed the checks it will be sent out for review. To provide the metadata for your submission, please Login to Editorial Manager (https://www.editorialmanager.com/pbiology) within two working days, i.e. by Oct 31 2025 11:59PM.

Kind regards,

Melissa

Melissa Vazquez Hernandez, Ph.D.

Associate Editor

PLOS Biology

---

## [Decision Letter · Decision Letter 1]

6 Jan 2026

Dear Jay,

Happy New Year!!

I hope you had a nice break. Thank you for your patience while your manuscript "Environmental redox conditions tune phenazine-mediated bacterial antagonism" was peer-reviewed at PLOS Biology. It has now been evaluated by the PLOS Biology editors, an Academic Editor with relevant expertise, and by three independent reviewers.

In light of the reviews, which you will find at the end of this email, we would like to invite you to revise the work to thoroughly address the reviewers' reports. As you will see below, the reviewers are positive about the relevance and novelty of the study, yet some concerns have raised during revision. Reviewer 1 requests additional experiments and controls to solidify the central mechanistic claims, particularly direct testing of purified phenazines under defined aerobic versus anaerobic conditions, validation of LC-MS peak identities with chemical standards, and experiments to reconcile the strong in vitro antagonism with the lack of Pa-mediated restriction of Kp in vivo and ex vivo. Reviewer 2 suggests to soften the claims about defining “interaction rules,” to expand environmental parameter testing if feasible, and to improve clarity of data presentation and figure organization. Reviewer 3 raises only minor, largely conceptual or presentation-focused points, including more cautious quantitative reporting, clarification of phenazine biosynthesis complexity, and contextualization of phenazine chemistry and concentrations, without calling for new experiments. We agree with the reviewer concerns and would require some additional experimental revisions to address them, as we consider that this would strengthen the work.

IMPORTANT: after discussion with the Academic Editor and the reviewers, whie we think that all experimental requests from Reviewer 1 will enhance the study, we would like the focus to be mainly on the direct testing of purified phenazines under defined aerobic versus anaerobic conditions. We would also encourage the validation of LC-MS peaks and the sequencing of the engineered strains. However, we would not ask for additional work to explain the difference between in vivo, ex vivo and in vitro.

Given the extent of revision needed, we cannot make a decision about publication until we have seen the revised manuscript and your response to the reviewers' comments. Your revised manuscript is likely to be sent for further evaluation by all or a subset of the reviewers.

**IMPORTANT - SUBMITTING YOUR REVISION**

*Re-submission Checklist*

*Published Peer Review*

*PLOS Data Policy*

*Blot and Gel Data Policy*

Sincerely,

Melissa

Melissa Vazquez Hernandez, Ph.D.

Associate Editor

PLOS Biology

REVIEWERS' COMMENTS

Reviewer #1:

This manuscript by Todd et al describes a novel interaction between P. aeruginosa isolates and Klebsiella isolates, in which Pa restricts the growth of Kp under aerobic conditions. The authors lead the reader through an initial observation from an unrelated mouse study in which oral gavage with Kp failed to establish gut colonization in mice, and sequencing of potential culprits revealed three Pa isolates capable of restricting Kp growth. The majority of the manuscript is devoted to probing the underlying mechanism, through which the authors discover that the growth-restricting factor is secreted and that restriction can be influenced by nutrient and oxygen availability. Through screening a transposon mutant library of a Pa strain, the authors determined that growth inhibition was likely due to phenazine production and regulation of production by quorum sensing. Through testing a series of Pa mutants and Ec heterologous expression constructs, the authors conclude that Pa largely restricts Kp growth through production of pyocyanin and pyorubin. Despite the initial mouse growth restriction data that prompted this study, colonization of mice by Pa failed to restrict Kp growth in vivo or in large intestinal content ex vivo. The authors conclude the body of work by demonstrating that a wide array of Klebsiella isolates and two E. coli isolates are restricted by phenazines, but Pa isolates vary widely in ability to restrict Kp growth. They also found that Pa isolates from different body sites differ in restrictive ability, with those from blood and urine being the most restrictive. Taken together, the authors state that these data provide evidence for environment-dependent modulation of pathogen interactions.

This is an enormous body of work, and provides further support for considering the genomic and environmental context of bacterial isolates with examining pathogen interactions. There are also several notable strengths, such as the assessment of strain-specific differences and directly quantifying PYO production by the clinical isolates and Pa strains and reconciling these data with the MIC for PYO against Kp. The manuscript is well-written overall, although there are some areas in the results section that are difficult to follow. The main weakness of the study is that the majority of the conclusions regarding the specific contributions of phenazines to growth restriction are only examined using spent media, without confirmation using purchased/purified PCA, PYO, 1-HP, etc. Another key weakness is the lack of Pa-mediated growth restriction in the mouse studies, although the authors address potential reasons for this discrepancy in the discussion. Further comments for consideration are provided below.

Major Points:

1. PCA, PYO, and 1-HP can all be purchased, yet the methods indicate that only PYO was purchased and it was only used for the MIC/MBC experiments. Direct confirmation of the ability of each chemical to restrict Kp growth in fresh media compared to spent media and under aerobic vs anaerobic conditions would substantially increase impact of the study and confidence in the conclusions.

2. In the discussion, the authors hypothesize that "environmental redox gates the potency of phenazine-mediated interference." This could be tested fairly easily by examining whether Kp growth is more highly restricted by purchased/purified phenazines under aerobic vs anaerobic conditions, and the results of such an experiment would increase the strength of the conclusions and overall impact of the body of work.

3. With respect to the LC-MS data, running purified PCA, PYO, and 1-HP as a standard would greatly increase confidence in the results. This is especially important as the peak that is assumed to be PCA for MG1655pPhzA-G eluted at a different time that the peak for PA14. Can any further information be provided to demonstrate level of confidence that this peak represents PCA, or whether the PCA produced by E. coli may differ in some way from that produced by PA14?

4. It is interesting that the supernatant from the PYR over-expressing strain was bactericidal, yet the PA14phzS mutant did not restrict growth of Kp, considering that inability to convert 5MPCA to PYO would be expected to result in accumulation of 5MPCA and PYR. Further comment on this observation is warranted in the discussion, particularly if it may be due to the unusual peaks in the PYO LCMS for this strain that are absent in the other strains.

5. All of the data are reported as log fold change in CFUs at experimental endpoint versus the input. While it does allow all data to be compared even if there are day-to-day changes in starting inoculum, it would useful to show the actual CFU values for key experiments, at least as supplemental or supporting information.

6. It was unclear from the body of the manuscript if the GS*M was a known/existing mutation or spontaneously created in this study. Upon re-reading the methods section, it appears to have been selected based on pigment production. I recommend providing additional rationale for the approach that was used, as well as sequencing the E. coli strain harboring the GS*M plasmid and providing a genome comparison to E. coli with either the empty vector or the GSM plasmid to confirm that no other changes have occurred in this strain that may impact Kp growth.

7. Figure 7 is missing a critical control for all panels. Please include the log fold change of each species after culture in MG1655 spent medium as well as PA14 spent medium to help the reader compare sensitivity to phenazines overproduction. Alternatively, please verify sensitivity to purchased/purified phenazines. Similarly, in Figure S14A, it would be helpful to include inhibition by PA14 spent media on this graph to support the statement in the text that "JV69 was inhibitory to a comparative level as PA14" especially as there appears to be little either limited growth of Kp in this experiment or limited inhibition by JV69.

8. The observation that gut colonization by Pa 191.1 was unable to restrict colonization by KPPR1 is important given the robust restriction in vitro. The authors note that this could be due to poor availability of environmental oxygen, and describe the change in mouse vendor in the discussion. However, the results section seems to abruptly shift to testing a potential role for the intact gut microbiota under anaerobic conditions and then interactions in other body sites without further examining the discrepancy. It would have been interesting to see the outcome of the ex vivo large intestinal content experiment if conducted under aerobic conditions instead of anaerobic, or determining whether purified PYO or PYR could inhibit KPPR1 under ex vivo conditions. Either one of these experiments would be an important addition to the body of work to attempt to further reconcile differences between the in vivo and ex vivo studies with the in vitro studies and proposed role of oxygen versus other potential confounding factors. It would also be useful to note whether the Pa isolates produced phenazines under the ex vivo culture conditions, as lack of production could also explain the lack of Kp growth restriction.

9. While examining competition between isolates from different body sites adds breadth to this body of work, it also leaves the reader wanting to know more about potential strain differences. For example, since the Pa strains isolated from blood and urine were more restrictive to Kp growth, it would be useful to know whether these strains tended to produce higher levels of phenazines than those from other sources, or whether their genomes cluster more closely with highly restrictive isolates like PA14 than other Pa isolates.

Additional points:

1. More information is needed regarding the study from Supplemental 1A that provided the foundation for this body of work. For example, how/why was it hypothesized that it was a gut bacterium causing the decrease in Kp colonization, what was the rationale for selecting the three noted mice, and what lead to selection of the isolates that ultimately turned out to be Pa?

2. For the genome sequencing in Fig S1B, the text of the legend and manuscript state that these strains "were most similar to sequence type ST175" but also state that MSLT was performed. Were the strains ST175, or a different ST?

3. In the results section prior to discussing Fig 3, the authors state "We observed that several mutants in our screen appeared to have modified phenazine production. The psqA, rhlR, and tipA mutants did not produce high amounts of pyocyanin (PYO) and pyorubin (PYR)." Please provide the data supporting these statements.

4. It is interesting that co-culture of Kp with Pa phzH restricts growth, but the supernatant of phzH does not restrict growth (Fig 3D vs 3E). Further discussion on possible mechanism of contact-dependent restriction by this mutant is warranted.

5. Please add PYR production to the diagram in Figure 3C to help further orient the reader as to why disrupting either phzA-G or phzHMS also disrupts PYR production. It is also recommend that the labels used in figure 5B (having the respective overproduced phenazine noted for each spent medium) be applied to all other relevant figure panels to help orient the reader.

6. For the experiments described in Figure 4, the rationale for using the PA14pPhzA-G spent medium is unclear, considering that this medium should have PCA similar to the MG1655pPhzA-G culture supernatant. Doe the data then suggest that PYO synergizes with PCA or other components of the spent media? It is also unclear if these studies were conducted using purchased/synthesized PYO, or PYO from the MG1655 phzS mutant.

7. In Figure S10, not all of the strain symbols indicated in the legend can be visually detected in panels B-F.

8. In Figure S12, it would be helpful to show the "strain alone" growth data for all Kp strains in comparison to PA14 restriction.

9. In Figure S13A, please indicate what the red and orange horizontal lines indicate.

10. In Figure 6, were mice and organ homogenates also inoculated with KPPR1 alone? If so, the addition of these data to panels B and C would be beneficial.

11. Pa CFUs are not reported in any of the figures, with the exception of one supplemental figure. Do any of the co-culture conditions that lack Kp growth restriction exhibit differences in Pa growth?

12. Please ensure that bacterial names are italicized, including when abbreviated (for example, the first paragraph of the Discussion has several references to Ec and Pa that are not italicized).

Reviewer #2:

This is a rigorous and very interesting study that addresses the role of phenazines in interbacterial competition. The study starts from the observation that several mice colonised with Pa show low colonisation with Kp. Restriction of Kp by Pa is investigated in depth and is shown to depend primarily on production of phenazines by Pa. Competitive outcomes vary depending on strain background and environmental conditions, and the authors present ex-vivo data suggesting that Pa-Kp antagonism is likely to vary by host niche. Overall I found the study to be thorough and well-presented and believe it will be of broad interest. The study also fills an important gap in the field by exploring interbacterial competition in mechanistic depth - although the picture it presents is very complicated, this type of study is very much needed and will eventually lead to the ability to predict competitive outcomes. I also commend the authors for their full description and thoughtful treatment of some of the unexpected results in this study (ie. that phenazine-based restriction of Kp by Pa is unlikely to occur in the mouse gut although this was suggested by the first experiment of the study).

I have only minor suggestions to improve the manuscript.

1. In the discussion I find the idea that this study defines "interaction rules" for phenazine-mediated competition too expansive for the data presented. The authors could broaden the experimental investigation into the effect of environment on phenazine-mediated competition in vitro (by examining how competition changes with added antioxidants, graded oxygen content in addition to oxic/anoxic, and presence of electron acceptors). If this is not feasible then I suggest this statement should be softened.

2. Introduction - The part of the paragraph starting "Redox-active natural products mediate…" should include some references.

3. Text presentation of Figure 1D - This is confusing to present ratios on ratios, it could be simpler to compare all of the growth rates to those in fresh LB?

4. Section on nutrient depletion - suggest making it explicit that these are single-strain growth experiments.

5. The section on phenazines and their contribution to restriction of Kp is difficult to follow at times. I suggest adding a small panel to Figure 3 to state which molecules are produced by which mutant or overexpression strain (although readers can piece this together from the overview of the biosynthesis), and stating which phenazines are not responsible for restriction when relevant (eg. when discussing lack of effect of the ∆phzH mutant).

6. Fix grammar in the sentence "Consistent with previous reports, the addition of reducing agent (generating reduced 5MPCA) is more active against Kp."

7. Section on oxygen dependence of Pa restriction of Kp: it would be very interesting to know whether Kp is undergoing anaerobic respiration or fermentation in these experiments - could the authors compare restriction with and without alternative electron acceptors? Alternatively, could they comment on which strategy is more likely based on the experimental conditions?

8. The reconciliation of the initial gut colonisation observations with further experiments is particularly thoughtful and thorough.

Reviewer #3 (Wulf Blankenfeldt):

The manuscript

Environmental redox conditions tune phenazine-mediated bacterial antagonism

by Todd, Schneider and colleagues deciphers the antagonistic activity of clinical isolates of Pseudomonas aeruginosa against Klebsiella pneumoniae. The work described in this manuscript is based on a different study from the authors' laboratory in which they have observed this antagonism caused by a kanamycin and rifampicin resistant bacterium, then identified as three different strains of P. aeruginosa.

The authors then follow with a convincing cascade of experiments, showing that the antagonism is triggered by phenazine derivatives synthesized by P. aeruginosa (namely pyocyanin and the lesser-known pyorubin(s)). They also show that this is not a "black-and-white" finding, but that the ability of P. aeruginosa to kill/stall K. pneumoniae depends on the environmental conditions, probably requiring the availability of oxygen and the absence of substances that can alleviate oxidative stress. Further, they demonstrate that not all P. aeruginosa strains can antagonize Klebsiella to the same extent. They also provide evidence that gram-negatives seem to be generally more sensitive to the P. aeruginosa phenazines than gram-positive bacteria.

The manuscript is well written and I enjoyed following the authors' story. I have only very minor remarks to make since I cannot really assess the quality of the experimental data because I work in a different research field:

- In the sentence "The growth of Kp strain KPPR1 was restricted 99.4%, 98.7%, and 97.5% by Pa 145.1, 191.1, and 193.1, respectively (Figure 1A)." on page 15 of the review document, I was wondering if the percentage numbers are stated a little too precisely and if they should not have standard deviations as well.

- On page 22 (analysis of transposon mutants), I was surprised that the authors have not also identified PqsE as one of the required factors, since it is quite well established that PqsE (the PQS response protein) is important/required for phenazine biosynthesis in P. aeruginosa.

- On the same page, my feeling was that the authors simplify the situation around phenazine biosynthesis in P. aeruginosa somewhat: P. aeruginosa has two phz-operons that will produce phenazine-1-carboxylic acid, the substrate for pyocyanin- and pyorubin-producing enzymes. There is no specification if they found transposon mutations in both operons.

- On page 23, the authors state that "Due to the chemical similarity between PYO and other phenazines, it is commonly assumed that these compounds exert similar biological effects." This may be correct, but it is quite well established in the literature that different phenazines have different redox potential. In addition, there are now several examples of phenazines not acting via their redox activity but through other modes of action (for example this one: https://pubmed.ncbi.nlm.nih.gov/40935925/)

- On page 27, the authors report the amount of pyocyanin that different strains of P. aeruginosa produce. I wonder if these numbers can be correct. For example, if there were indeed 20.7 mg/ml from P. aeruginosa PA14, this would amount to 20.7 g per liter of culture, roughly 0.1 Mol. This sounds like an awful lot, but I may be mistaken. However, such high concentrations would indicate that pyocyanin/pyorubin are very ineffective bacteriostatics/antibiotics. In this context, the authors may want to read a little on "pyocyanase", an antibiotically active concoction from P. aeruginosa marketed in the pre-antibiotic era.

- Still at this same spot on page 27, I would also appreciate if the authors could state the concentrations also in molar units.

- On page 39, I was not sure what the authors mean by "true, true, unrelated".

---

## [Decision Letter · Decision Letter 2]

9 Apr 2026

Dear Jay,

Thank you for your patience while we considered your revised manuscript "Environmental redox conditions tune phenazine-mediated bacterial antagonism" for publication as a Research Article at PLOS Biology. This revised version of your manuscript has been evaluated by the PLOS Biology editors, the Academic Editor and the original reviewers, who congratulate you in your great work!

Based on the reviews, we are likely to accept this manuscript for publication, provided you satisfactorily address the remaining editorial points. Please also make sure to address the following data and other policy-related requests.

1) We routinely suggest changes to titles to ensure maximum accessibility for a broad, non-specialist readership, and to ensure they reflect the contents of the paper. In this case, we would suggest a minor edit to the title, as follows. Please ensure you change both the manuscript file and the online submission system, as they need to match for final acceptance:

"Environmental redox conditions and strain variation define phenazine-mediated antagonism in co-infecting bacteria"

Please supply the numerical values either in the a supplementary file or as a permanent DOI’d deposition for the following figures:

Figure 1A-G, 2A-D, 3ABDEF, 4A-H, 5A-E, 6ABC, 7A-E; S1A, S2A-D, S3A-D, S4A-D, S5A-J, S6AB, S7B-F, S8A-D, S9, S11A-F, S12ABC, S13AB, S14AB, S15A, S16A-D, S17ABC

*I am aware that you have provided the raw data both in Github and also in our system. However, it is not clear to what data they belong to or what scripts to use for each figure. Could you provide perhaps a sheet specifying this?

3) Please cite the location of the data clearly in all relevant main and supplementary Figure legends, e.g. “The data underlying this Figure can be found in S1 Data” or “The data underlying this Figure can be found in https://doi.org/10.5281/zenodo.XXXXX”

4) For figures containing any spectrometry data (Figures 3G, S10AB, S15B), please deposit the data on publicly available databases (see suggested db here; https://journals.plos.org/plosbiology/s/recommended-repositories) and provide the accession number/URL of the deposition in the Data Availability Statement in the online submission form.

5) Please provide the tree files for the phylogenetic trees in Figures S1B. Please make sure all relevant figures have scale bars.

6) Supplementary files (e.g., excel). Please ensure that all data files are uploaded as 'Supporting Information' and are invariably referred to (in the manuscript, figure legends, and the Description field when uploading your files) using the following format verbatim: S1 Data, S2 Data, etc. Multiple panels of a single or even several figures can be included as multiple sheets in one excel file that is saved using exactly the following convention: S1_Data.xlsx (using an underscore).

7) Please ensure that your Data Statement in the submission system accurately describes where your data can be found and is in final format, as it will be published as written there

8) Thank you for providing the underlying code in GitHub. However, because Github depositions can be readily changed or deleted, please make a permanent DOI’d copy (e.g. in Zenodo) and provide this URL in the manuscript and Data Availability Statement.

We expect to receive your revised manuscript within two weeks.

*Published Peer Review History*

*Press*

Sincerely,

Melissa

Melissa Vazquez Hernandez, Ph.D.

Associate Editor

PLOS Biology

REVIEWERS' COMMENTS

Reviewer #1: The authors have done a commendable job addressing all reviewer comments.

Reviewer #2: The authors have addressed all of my suggestions and I have no further concerns or changes. Congratulations on a great study.

Reviewer #3 (Wulf Blankenfeldt): Todd, Schneider and colleagues have submitted a thoroughly revised version of their manuscript

Environmental redox conditions tune phenazine-mediated bacterial antagonism.

I am very happy to see that they have worked hard to satisfy the revision requests handed in by two colleagues and me. Because I work less in the biological field myself and can hence not really comment on the remarks provided by my referee colleagues, my own questions and corrections have all been answered such that I do not have further points to criticize.

---

## [Editor Report · Decision Letter 3]

5 May 2026

Dear Jay,

Thank you for the submission of your revised Research Article "Environmental redox conditions and strain variation define phenazine-mediated antagonism in co-infecting bacteria" for publication in PLOS Biology. On behalf of my colleagues and the Academic Editor, Sara Mitri, I am pleased to say that we can in principle accept your manuscript for publication, provided you address any remaining formatting and reporting issues. These will be detailed in an email you should receive within 2-3 business days from our colleagues in the journal operations team; no action is required from you until then. Please note that we will not be able to formally accept your manuscript and schedule it for publication until you have completed any requested changes.

PRESS

Sincerely,

Melissa

Melissa Vazquez Hernandez, Ph.D., Ph.D.

Associate Editor

PLOS Biology
